# Co-registration and residual correction of digital elevation models: A comparative study

Tao Li[1,2], Yuanlin Hu[1,3], Bin Liu[4], Liming Jiang[1,2], Hansheng Wang[1], Xiang Shen[1]

[1]State Key Laboratory of Geodesy and Earth's Dynamics, Innovation Academy for Precision Measurement Science and Technology, Chinese Academy of Sciences, Wuhan, Hubei, 430077, China.
[2]University of Chinese Academy of Sciences, Beijing 100049, China.
[3]School of Geography and Information Engineering, China University of Geosciences, Wuhan, Hubei, 430074, China.
[4]State Key Laboratory of Remote Sensing Science, Aerospace Information Research Institute, Chinese Academy of Sciences, Beijing 100101, China.

*Correspondence to*: Xiang Shen (shen@apm.ac.cn)

## Abstract

Digital elevation models (DEMs) are currently one of the most widely used data sources in glacier thickness change research, due to the high spatial resolution and continuous coverage. However, raw DEM data are often misaligned with each other, due to georeferencing errors, and a co-registration procedure is required before DEM differencing. In this paper, we present a comparative analysis of the two classical co-registration methods proposed by Nuth and Kääb (2011) and Rosenholm and Torlegard (1988). The former is currently the most commonly used method in glacial studies, while the latter is a seminal work in the photogrammetric field that has not been extensively investigated by the cryosphere community. Furthermore, we also present a new residual correction method using a generalized additive model (GAM) to eliminate the remaining systematic errors in DEM co-registration results. The performance of the two DEM co-registration methods and three residual correction algorithms (the GAM-based method together with two parametric-model-based methods) was evaluated using multiple DEM pairs from the Greenland Ice Sheet and mountain glaciers, including Advanced Spaceborne Thermal Emission and Reflection Radiometer (ASTER) DEMs, ZiYuan-3 (ZY-3) DEMs, the Shuttle Radar Topography Mission (SRTM) DEM, and the Copernicus DEM. The experimental results confirm our theoretical analysis of the two co-registration methods. The method of Rosenholm and Torlegard has a greater ability to remove DEM misalignments (an average of 4.6% and 13.7% for the test datasets from Greenland Ice Sheet and High Mountain Asia, respectively) because it models the translation, scale, and rotation-induced biases, while the method of Nuth and Kääb considers translation only. The proposed GAM-based method performs statistically better than the two residual correction methods based on parametric regression models (high-order polynomials and the sum of the sinusoidal functions). A visual inspection reveals that the GAM-based method, as a non-parametric regression technique, can capture complex systematic errors in the DEM co-registration residuals.

## 1. Introduction

Differencing between multi-temporal digital elevation models (DEMs) is a widely used approach for mapping glacier elevation changes at local and regional scales (Bolch et al., 2011; Gardelle et al., 2013; Pieczonka et al., 2013; Liu et al., 2019; Hugonnet et al., 2021). However, limited by the imaging and georeferencing techniques, systematic errors often exist in the raw DEMs (Rodriguez et al., 2006), which can lead to wrong estimation of glacier mass change and false detection of glacier surges (Nuth and Kääb, 2011). Numerous studies have confirmed that a co-registration process is required to remove these biases before DEM differencing is conducted (Van Niel et al., 2008; Nuth and Kääb, 2011; Paul et al., 2015).

DEM co-registration has been extensively studied, and the existing methods can be broadly classified into two main categories. The first category requires an explicit data matching process (i.e., correspondence search). Typical methods in this category include: feature point based methods, e.g., the scale-invariant feature transform (SIFT) descriptor (Aguilar et al., 2012; Sedaghat and Naeini, 2018) and the method based on centroids of subwatersheds (Li et al., 2017); feature line based methods, e.g., methods based on stream networks or watershed boundaries (Karkee et al., 2008); the multi-feature based surface matching method (Wu et al., 2013); the iterative closest point (ICP) algorithm and its variants (Besl and Mckay, 1992; Rusinkiewicz and Levoy, 2001; Di et al., 2012); and the least squares 3D surface matching (LS3D) algorithm (Gruen and Akca, 2005; Akca, 2010). All of the above methods originate from image or point cloud processing studies, and they can be used for the coarse co-registration of DEMs without georeferenced information. However, the main disadvantage of these methods is that the correspondence finding procedure is very time-consuming when processing large DEMs. Moreover, the accuracy of the image-based methods (e.g., SIFT) is strongly dependent on extracting a large number of high-quality features, which is not an easy task for DEMs lacking sufficient textures.

The second category of DEM co-registration methods does not require an explicit matching process. The optimization objective of these methods is usually to minimize the sum of the vertical distances between two DEMs, where each pixel in the secondary DEM implicitly corresponds to the same planimetric position in the reference DEM. These methods are not suitable for images lacking georeferenced information, but they are strongly recommended for high-accuracy applications where the DEMs have been georeferenced or coarsely co-registered. The typical algorithms in this category include grid search methods (Hofton et al., 2006; Berthier et al., 2007; Van Niel et al., 2008; Cucchiaro et al., 2020) and terrain information based methods (Gorokhovich and Voustianiouk, 2006; Peduzzi et al., 2010; Nuth and Kääb, 2011). The grid search methods search for the best alignment result by stepwise shifting the secondary DEM a little bit, alternatively along x and y directions in a predefined window (e.g., 5 × 5 pixels). However, these methods have been rarely used in the recent literature because their brute-force search process comes with a huge computational cost. The terrain information based methods are derived from the analytical relationship between the elevation differences of the DEMs and terrain-related information. The method proposed by Nuth and Kääb (2011) employs the terrain slope and aspect as explanatory variables in the regression model, and is currently the most commonly used DEM co-registration algorithm in glacial studies (Vacaflor et al., 2022). In a much earlier study, Rosenholm and Torlegard (1988) developed an absolute orientation algorithm for stereo models based on terrain gradients.

This method has been widely used for DEM co-registration in the photogrammetry field, but, unfortunately, has rarely been considered in the cryosphere community.

The goal of this paper is two-fold. The first goal is to reveal the connections and differences between the slope/aspect based method of Nuth and Kääb (2011) (hereinafter referred to as NK) and the terrain gradient based DEM co-registration algorithm of Rosenholm and Torlegard (1988) (hereinafter referred to as RT), and the second goal is to present a non-parametric approach to remove the complex systematic errors in DEM co-registration results.

## 2. DEM co-registration

This section focuses on the analytical (i.e., the terrain information based) DEM co-registration methods only. We will demonstrate that the NK and RT methods are theoretically compatible. As the original algorithms in the works of NK and RT were presented in distinct forms, we will present detailed derivations of the equations used in their algorithms and variants.

### 2.1. The method of Nuth and Kääb

### 2.1.1. Standard version

The equations of the NK method are derived from the geometric relationship (cf. Fig. 1) of the elevation differences induced by the DEM shift with respect to the terrain slope ($\theta$) and aspect ($\psi$) values. Firstly, we consider the special case where $b = \psi$ (where $b$ is the aspect of the shift vector), i.e., the translation is exactly along the terrain aspect direction. As shown in Fig. 1b, the induced elevation difference is given by:

$$\begin{aligned}
dH &= dH_{XY} + dH_Z \\
&= \text{FE} + \text{EG} \\
&= \text{OE} \cdot \tan(\theta) + \text{EG} \\
&= a \cdot \tan(\theta) + c
\end{aligned} \tag{1}$$

where $a$ and $c$ are the horizontal and vertical distances of the shift vector, respectively.

In a more general scenario, $b \neq \psi$. As shown in Fig. 1a, the horizontal shift vector **OE'** is decomposed into **OE** and **EE'**. Since **EE'** is perpendicular to the vertical plane OEF defined by the gradient vector and the terrain aspect direction, this does not cause any elevation change. The vertical difference induced by **OE'** is therefore equal to that of **OE**, and it exists:

$$\begin{aligned}
dH &= \text{FE} + \text{EG} \\
&= \text{OE} \cdot \tan(\theta) + \text{EG} \\
&= \text{OE}' \cdot \cos(b - \psi)\tan(\theta) + \text{E}'\text{G}' \\
&= a \cdot \cos(b - \psi)\tan(\theta) + c
\end{aligned} \tag{2}$$

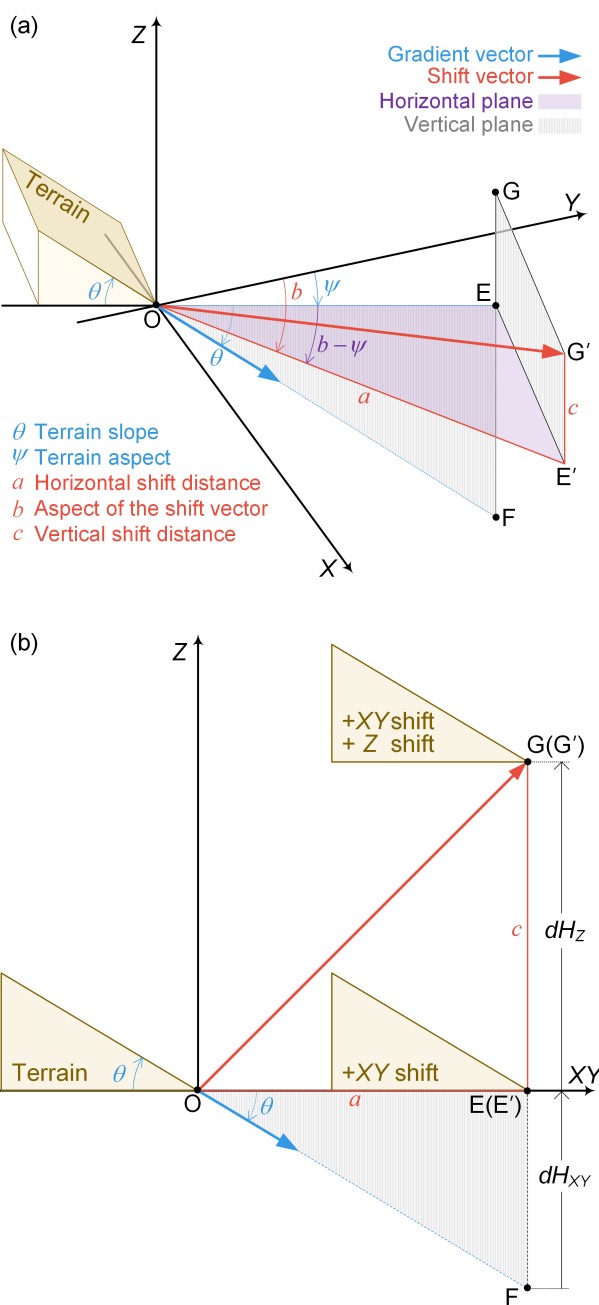

**Figure 1.** Elevation differences induced by DEM shift. **(a)** 3-D view when $b \neq \psi$. **(b)** 2-D view when $b = \psi$.

The above is Equation (2) in Nuth and Kääb (2011). In this paper, we refer to this as the NK standard version. The cylindrical coordinates $(a, b, c)$ of the shift vector can be estimated from a nonlinear regression of Eq. (2), and the corresponding Cartesian coordinates are then given by:

$$\Delta X = a \cdot \sin(b)$$
$$\Delta Y = a \cdot \cos(b) \tag{3}$$
$$\Delta Z = c$$

As shown in Fig. 2, an iterative process is generally required for accurate co-registration of two DEMs (Nuth and Kääb, 2011), where the coordinates of the secondary DEM are updated in every iteration by the following equation:

$$\begin{bmatrix} X \\ Y \\ Z \end{bmatrix}_i = \begin{bmatrix} X \\ Y \\ Z \end{bmatrix}_{i-1} + \begin{bmatrix} \Delta X \\ \Delta Y \\ \Delta Z \end{bmatrix} \tag{4}$$

where the subscript $i$ represents the $i$-th iteration. The iterative process terminates when the change in the dispersion characteristics (median absolute deviation from zero) of the elevation differences between iterations is less than a predefined threshold.

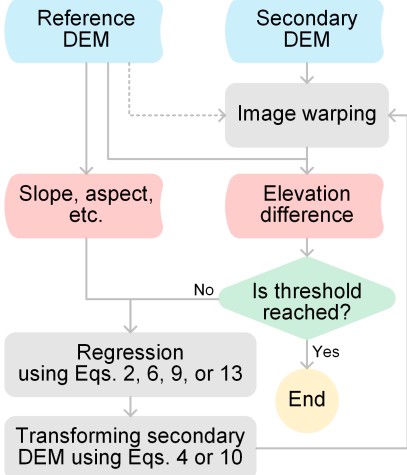

**Figure 2.** DEM co-registration flowchart.

## 2.1.2. Simplified version

Nuth and Kääb (2011) did not use Eq. (2) in their experiments, but instead adopted a simplified regression equation by dropping one explanatory variable ($\theta$). Firstly, both sides of Eq. (2) are divided by $\tan(\theta)$:

$$\frac{dH}{\tan(\theta)} = a \cdot \cos(b - \psi) + \frac{c}{\tan(\theta)} \tag{5}$$

$\theta$ in the right side of the equation is then approximately replaced by the mean terrain slope of the DEM:

$$\frac{dH}{\tan(\theta)} \approx a \cdot \cos(b - \psi) + c' \tag{6}$$

where

$$c' = \frac{c}{\tan\left(\mathrm{mean}\left(\theta\right)\right)} \tag{7}$$

Accordingly, the shift vector in the Cartesian coordinate system is given by:

$$\Delta X = a \cdot \sin\left(b\right)$$
$$\Delta Y = a \cdot \cos\left(b\right) \tag{8}$$
$$\Delta Z = c' \cdot \tan\left(\mathrm{mean}\left(\theta\right)\right)$$

### 2.1.3. Linear version

The standard version of the NK method can be converted to a linear regression equation by combining Eq. (2) with Eq. (3):

$$\begin{aligned} dH &= a \cdot \cos\left(b - \psi\right)\tan\left(\theta\right) + c \\ &= a \cdot \left(\sin\left(b\right)\sin\left(\psi\right) + \cos\left(b\right)\cos\left(\psi\right)\right)\tan\left(\theta\right) + c \\ &= \sin\left(\psi\right)\tan\left(\theta\right)\Delta X + \cos\left(\psi\right)\tan\left(\theta\right)\Delta Y + \Delta Z \end{aligned} \tag{9}$$

This equation uses $\Delta X, \Delta Y$, and $\Delta Z$ as the regression coefficients directly, and, accordingly, the conversion of the shift vector from cylindrical coordinates to Cartesian coordinates is no longer required.

### 2.2. The method of Rosenholm and Torlegard

In the RT method, the misalignment between two DEMs is described by a 3-D similarity transformation (Molodenskii, 1962; Badekas, 1969), and the coordinate update equation for the secondary DEM is:

$$\begin{bmatrix} X_C \\ Y_C \\ Z_C \end{bmatrix}_i = \left(1 + \gamma\right) \begin{bmatrix} 1 & -\kappa & \varphi \\ \kappa & 1 & -\omega \\ -\varphi & \omega & 1 \end{bmatrix} \begin{bmatrix} X_C \\ Y_C \\ Z_C \end{bmatrix}_{i-1} + \begin{bmatrix} \Delta X \\ \Delta Y \\ \Delta Z \end{bmatrix} \tag{10}$$

where $\gamma$ is the scale factor; $\omega, \varphi$, and $\kappa$ are the rotation angles (in radians) about the $X$, $Y$, and $Z$ axes, respectively; and the

subscript C refers to the coordinates being zero-centered. Note that the values of $\gamma, \omega, \varphi$, and $\kappa$ are relatively small in the DEM co-registration process. The coordinate changes in each iteration can be approximated as:

$$\begin{bmatrix} X_C \\ Y_C \\ Z_C \end{bmatrix}_i - \begin{bmatrix} X_C \\ Y_C \\ Z_C \end{bmatrix}_{i-1} \approx \begin{bmatrix} \Delta X + \gamma X_C - \kappa Y_C + \varphi Z_C \\ \Delta Y + \kappa X_C + \gamma Y_C - \omega Z_C \\ \Delta Z - \varphi X_C + \omega Y_C + \gamma Z_C \end{bmatrix} \tag{11}$$

By comparing Eq. (11) with Eq. (4), it can be seen that the coordinate change of every DEM pixel is a constant vector ($\Delta X, \Delta Y, \Delta Z$) in the NK method, while it varies with the position ($X_C, Y_C, Z_C$) in the RT method. Accordingly, the following

equation is derived by substituting Eq. (11) into Eq. (9):

$$dH = \sin\left(\psi\right)\tan\left(\theta\right)\left(\Delta X + \gamma X_C - \kappa Y_C + \varphi Z_C\right) + \cos\left(\psi\right)\tan\left(\theta\right)\left(\Delta Y + \kappa X_C + \gamma Y_C - \omega Z_C\right) + \left(\Delta Z - \varphi X_C + \omega Y_C + \gamma Z_C\right) \tag{12}$$

By rearranging the above equation, the DEM elevation differences caused by translation, scaling, and rotation can be obtained as:

$$dH = v_{\Delta X}\Delta X + v_{\Delta Y}\Delta Y + \Delta Z + v_{\gamma}\gamma + v_{\omega}\omega + v_{\varphi}\varphi + v_{\kappa}\kappa \tag{13}$$

where

$$
\begin{aligned}
v_{\Delta X} &= \sin(\psi)\tan(\theta) \\
v_{\Delta Y} &= \cos(\psi)\tan(\theta) \\
v_{\gamma} &= v_{\Delta X}X_{C} + v_{\Delta Y}Y_{C} + Z_{C} \\
v_{\omega} &= Y_{C} - v_{\Delta Y}Z_{C} \\
v_{\varphi} &= v_{\Delta X}Z_{C} - X_{C} \\
v_{\kappa} &= v_{\Delta Y}X_{C} - v_{\Delta X}Y_{C}
\end{aligned}
\tag{14}
$$

It can be found from the geoscience literature that the slope and aspect angles relate to the terrain gradients, and the following equation (Peckham and Jordan, 2007) exists when the terrain aspect is measured clockwise from north:

$$
\begin{aligned}
\theta &= \arctan\sqrt{f_X^2 + f_Y^2} \\
\psi &= \pi - \arctan\left(\frac{f_Y}{f_X}\right) + \frac{\pi}{2}\left(\frac{f_X}{|f_X|}\right)
\end{aligned}
\tag{15}
$$

where $f_X$ and $f_Y$ are the gradients of the terrain in the $X$ and $Y$ directions, respectively. From the above equation, we obtain:

$$
\begin{aligned}
f_X &= -\sin(\psi)\tan(\theta) \\
f_Y &= -\cos(\psi)\tan(\theta)
\end{aligned}
\tag{16}
$$

Finally, by substituting Eq. (16) into Eq. (14), Eq. (13) is as follows:

$$dH = -f_X\Delta X - f_Y\Delta Y + \Delta Z + v_{\gamma}\gamma + v_{\omega}\omega + v_{\varphi}\varphi + v_{\kappa}\kappa \tag{17}$$

The above equation is Equation (6) in Rosenholm and Torlegard (1988). Comparing Eqs. (17) and (13) with Eq. (9), it can be seen that the RT and NK methods are theoretically compatible with each other, though the algorithms in their original works were presented in distinct forms.

## 3. Residual correction

A residual correction procedure is highly recommended after DEM co-registration (Berthier et al., 2007; Girod et al., 2017) because some systematic errors related to the terrain height and satellite acquisition geometry (along-track and cross-track) often remain. As elevation-dependent biases were not observed in our experiments, the following section introduces the residual correction algorithms for the along-track and cross-track directions only.

### 3.1. Parametric regression

High-order polynomial (6th to 8th order) regression is the most commonly used way to fit DEM co-registration residuals (Nuth and Kääb, 2011; Gardelle et al., 2013; Berthier et al., 2016; Brun et al., 2017), and is usually performed in a stepwise manner:

$$dH_{X_t} = \sum_{i=0}^{m} P_i X_t^{i}$$
$$dH_{Y_t} = \sum_{j=0}^{m} P_j Y_t^{j} \tag{18}$$

with

$$X_t = X \cos(\theta_t) - Y \sin(\theta_t)$$
$$Y_t = X \sin(\theta_t) + Y \cos(\theta_t) \tag{19}$$

where $X_t$ and $Y_t$ are the cross-track and along-track coordinates, respectively; $\theta_t$ is the angle between the along-track direction and the north; $m$ is the degree of the polynomial; and $P_i$ and $P_j$ are the coefficients to be estimated.

Previous studies have reported that the residual signals in the along-track direction often appear at one to three frequencies, and are most likely induced by satellite attitude jitter, which is mainly caused by high-frequency mechanical vibration (Nuth and Kääb, 2011). Girod et al. (2017) pointed out that these periodic residuals can be modeled by a sum of the sinusoidal functions:

$$dH_{X_t} = \sum_{i=0}^{m} P_i X_t^{i}$$
$$dH_{Y_t} = \sum_{k=1}^{n} A_k \sin\left(2\pi f_k Y_t + \varphi_k\right) \tag{20}$$

where $n$ is the number of sinusoidal functions; and $A_k, f_k$, and $\varphi_k$ are the amplitude, frequency, and phase of the $k$-th sinusoidal component, respectively.

### 3.2. Non-parametric regression

We propose an alternative residual correction method using a generalized additive model (GAM):

$$dH = s\left(X_t\right) + s\left(Y_t\right) \tag{21}$$

where $s(*)$ represents a smooth function. As an extension of the linear model by including additive smooth functions for the explanatory variables, the GAM has the potential to capture complex nonlinear patterns that a parametric model (e.g., high-order polynomials and sinusoidal functions) would miss.

The GAM software packages are widely available in various programming languages, such as R, Python, Matlab, and SAS. Typical smooth functions include local polynomials, splines, Markov Random Fields, and Gaussian process smooths. In

our experiments, the GAM regression of Eq. (21) was performed in R software using the 'mgcv' package (Wood, 2022). A thin-plate spline was chosen as the smoothing basis (i.e., the smooth function $s$), and the degree of smoothing was automatically determined by the generalized cross validation (GCV) criterion. For more on the theoretical foundations and technical details of the GAM method and the 'mgcv' package, we refer the reader to Hastie and Tibshirani (1990) and Wood (2017).

## 4. Experiments

### 4.1. Ice Sheet case study

#### 4.1.1. Data processing

The comparative experiments of DEM co-registration and residual correction were carried out on 23 Advanced Spaceborne Thermal Emission and Reflection Radiometer (ASTER) DEM pairs from the western edge of the Greenland Ice Sheet (GrIS) (Fig. 3). Details of all ASTER DEM pairs are provided in the Supplement (Table S1), where two DEM pairs were used for visualization and analysis (Table 1). The raw stereoscopic DEMs were automatically produced by the US Geological Survey Land Processes Distributed Active Archive Center (LPDAAC) using SilcAst software (NASA et al., 2001).

The normalized difference bareness index (NDBI) was calculated from Landsat 8 images to extract stable regions (Nguyen et al., 2021):

$$NDBI = \frac{SWIR1 - G}{SWIR1 + G} \tag{22}$$

where SWIR1 and G represent the first shortwave infrared band (1.560–1.660 μm) and the green band (0.525–0.600 μm) of the Landsat 8 Operational Land Imager (OLI) data, respectively. All the terrain-related information (slope, aspect, etc.), which served as explanatory variables of the regression, was then derived from the reference DEMs. In the co-registration and residual correction procedures, only DEM pixels over stable terrain were used for the regression, and a three-sigma rule (i.e., more than three times the standard deviation) was employed on the elevation differences between two DEMs to remove erroneous data caused by misclassification of unstable terrain areas. A subset of the data (of no more than 50,000 pixels, to reduce the computational cost) was randomly selected as the training set, and the remaining pixels were used for the accuracy evaluation by comparing the median absolute difference (MedAD) (Mcmillan et al., 2019; Trevisani and Rocca, 2015):

$$MedAD = \text{median}\left(\left|H_{\text{Reference}} - H_{\text{Secondary}}\right|\right) \tag{23}$$

where $H_{\text{Reference}}$ and $H_{\text{Secondary}}$ represent the reference and secondary DEM elevation, respectively.

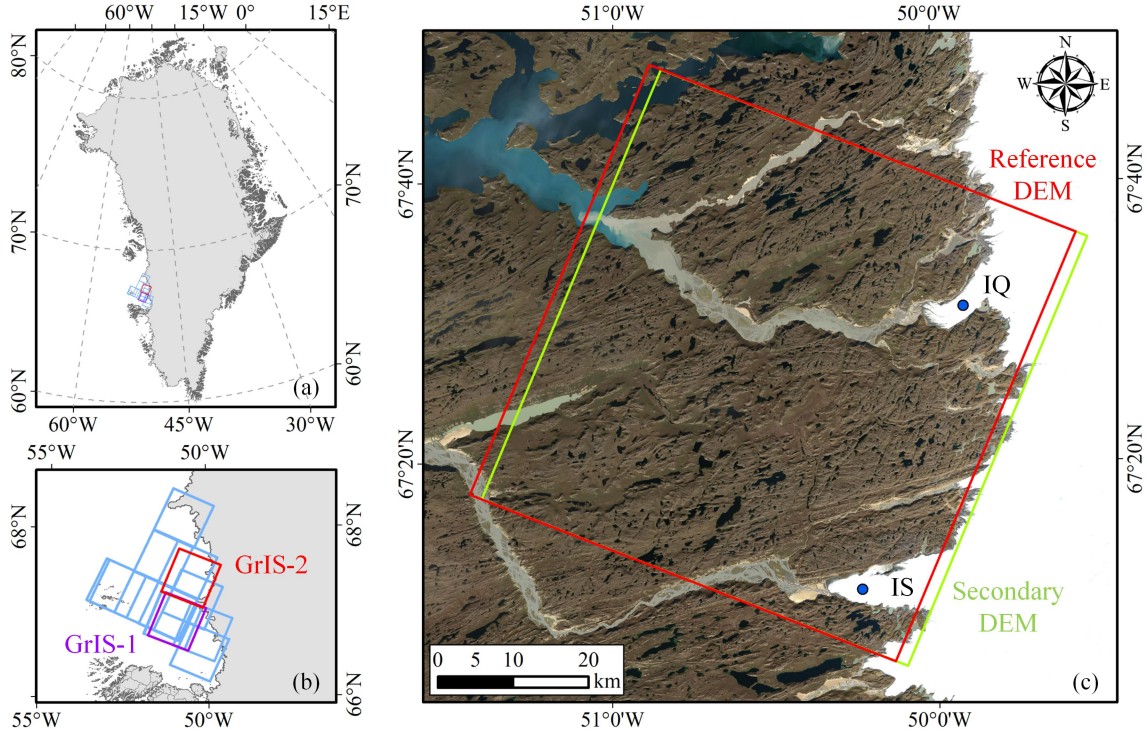

**Figure 3.** The study area located on the western edge of GrIS. **(a)** and **(b)** The footprints (blue) of the 23 ASTER DEM pairs, where pairs GrIS-1 and GrIS-2 listed in **Table 1** are highlighted in purple and red, respectively. The ice-covered areas are shown in gray, and their outlines are from Rignot and Mouginot (2012) and the Randolph Glacier Inventory (RGI) 6.0 (Pfeffer et al., 2014; Rastner et al., 2012; RGI Consortium, 2017) for the GrIS and its peripheral glaciers (the glaciers with connectivity level 2 are excluded), respectively. **(c)** The coverage of the two DEM images in pair GrIS-2 (red: reference DEM; green: secondary DEM). IQ and IS are the Inugpait Quat Glacier and Isunguata Sermia Glacier, respectively. The background image was acquired by Landsat 8 in 2016.

**Table 1.** Characteristics of the two DEM pairs in GrIS.

| Pair ID | Roles | Date | Res. (m) | Scene ID |
|---------|-------|------|----------|----------|
| GrIS-1 | Reference DEM | 5 Aug 2014 | 30 | AST14DEM.003:2133338256 |
| | Secondary DEM | 7 Aug 2003 | 30 | AST14DEM.003:2015893657 |
| GrIS-2 | Reference DEM | 25 Jul 2016 | 30 | AST14DEM.003:2237110490 |
| | Secondary DEM | 17 Jun 2002 | 30 | AST14DEM.003:2007321075 |

### 4.1.2. DEM co-registration

Table 2 shows that all four co-registration methods effectively reduce the DEM biases, and the following findings were made by comparing the error statistics of the different algorithms.

1) The standard and linear versions of the NK method yield exactly the same outcomes. The only difference between the two algorithms is whether the regression equation is linear or not, which does not affect the co-registration results.

2) The NK simplified version produces similar results to the standard version. It should be noted that this conclusion may not hold true for other datasets, because it cannot be proven theoretically that approximating terrain slopes by their mean value would always lead to a reliable performance.

3) The RT method performs better than the three versions of the NK method. The co-registration errors of the RT method are smaller than those of the NK linear algorithm by an average of 4.6% and a maximum of 15.3%, which indicates that there

are some scale- and rotation- induced biases in the experimental DEM data.

**Table 2.** Co-registration results obtained with the 23 DEM pairs of GrIS.

| Method | Average MedAD (m) |
|---|---|
| Before co-registration | 12.043 |
| NK standard version | 7.170 |
| NK simplified version | 7.163 |
| NK linear version | 7.170 |
| RT | 6.839 |

Figure 4 shows the elevation differences of DEM pair GrIS-1 before co-registration. All the pixels classified as water and potential outliers due to clouds were masked out for a better visualization, leaving the regions of bare land and glacier

(bounded by the black lines). It can be seen from the figure that most pixels are negative values, indicating that the majority of the elevation differences are caused by vertical translation. Minor errors related to the terrain (induced by horizontal translation) and along-track coordinates (caused by jitter) can also be clearly observed.

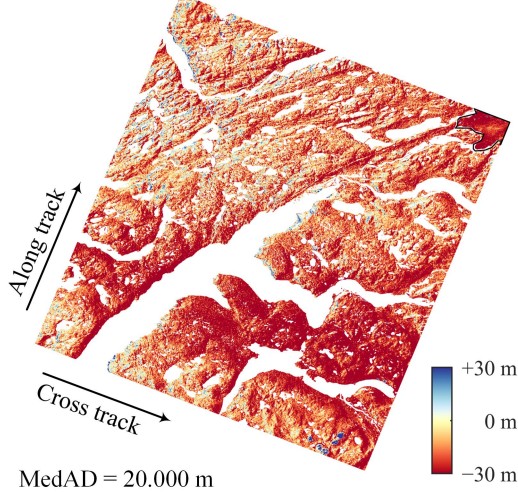

**Figure 4.** The elevation differences before DEM co-registration (pair GrIS-1). The black lines mark the GrIS boundaries delineated by
Rignot and Mouginot (2012).

The elevation difference maps (Fig. 5) demonstrate that the residuals of all three versions of the NK co-registration algorithms are consistent in terms of both magnitude and distribution. The RT algorithm shows better co-registration results, with removing 11.8% more errors compared to the NK linear version. A visual inspection of Fig. 5 c and d reveals that the elevation differences of NK exhibit a positive trend in the northwest corner (the blue circle in Fig. 5c) and a negative trend in the southeast corner (the red circle in Fig. 5c), which are possibly caused by unconsidered attitude biases. In addition, some clustered outliers, which may consist of misclassified water and cloud pixels, can be clearly observed in the elevation difference maps (Fig. 5a). However, these outliers have little influence on the co-registration results because robust statistical methods (robust regression algorithms and a robust scale estimation method, i.e., MedAD) were used in the experiments.

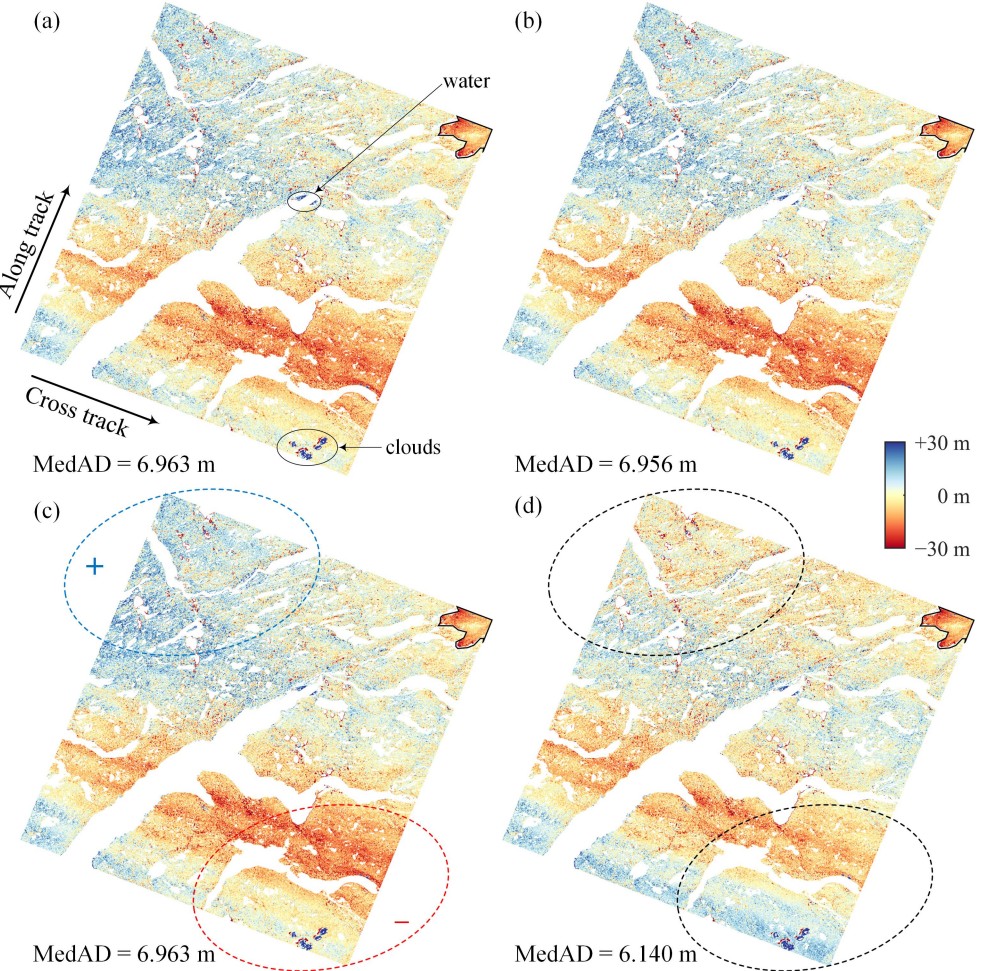

**Figure 5.** Co-registration results of the different methods for DEM pair GrIS-1: the standard **(a)**, simplified **(b)**, and linear **(c)** versions of the NK method, and the RT method **(d)**.

### 4.1.3. Residual correction

The residual correction results for DEM pair GrIS-1 are shown in Fig. 6. In the experiments, the polynomial fitting method used an 8th-order polynomial sequentially in the cross-track and along-track directions, and the combination of polynomial and the sum of sines method was implemented by first adopting an 8th-order polynomial in the cross-track direction and then applying a sum of three sines in the along-track direction. A visual comparison reveals that the high-order polynomial removes the low-frequency residuals only, whereas both the sum of sines and the GAM spline can capture the high-frequency signals. The MedAD values show that the GAM spline fitting method yields a higher accuracy than the two parametric regression methods.

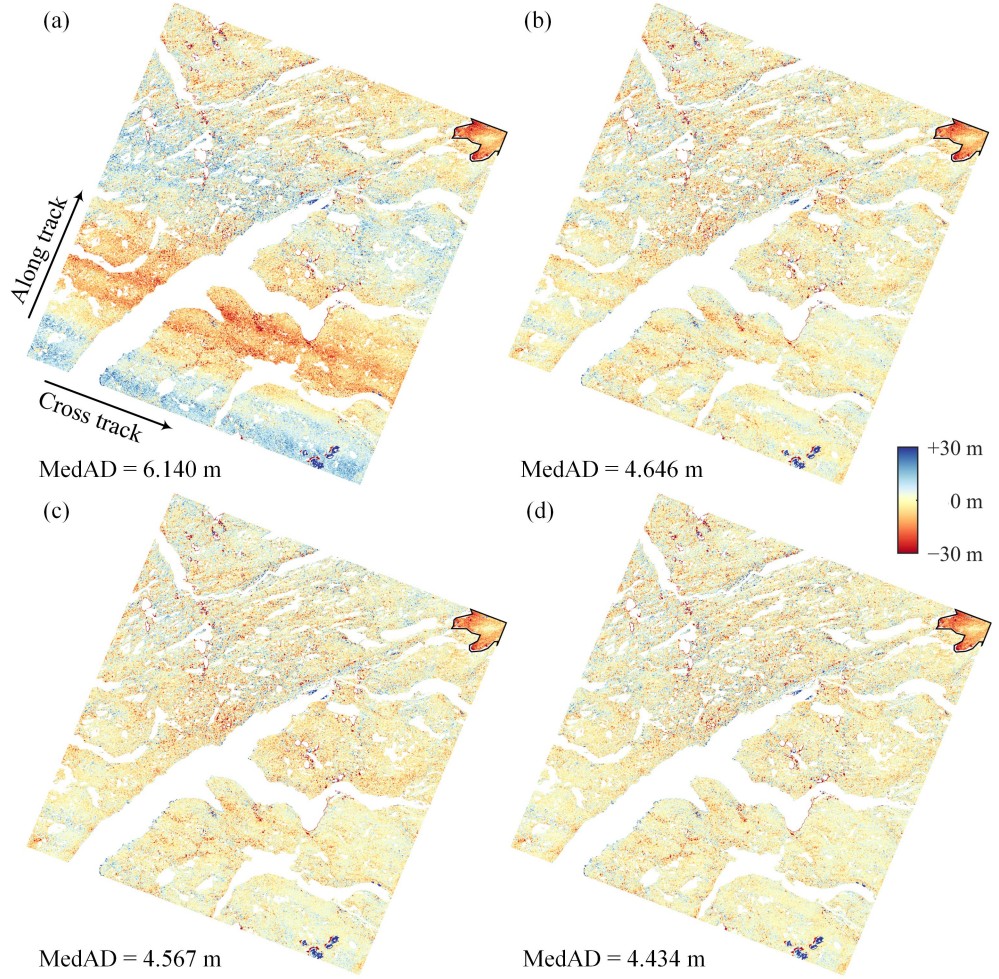

**Figure 6.** Residual correction of DEM pair GrIS-1. **(a)** The DEM co-registration results obtained using the RT method. The residual correction results obtained using polynomial fitting **(b)**, the combination of polynomial and the sum of sines method **(c)**, and GAM spline fitting **(d)**.

The magnitude of the high-frequency signals in DEM pair GrIS-2 is much greater than that in DEM pair GrIS-1 in Fig. 7b. The polynomial fitting method again eliminates only the low-frequency residuals. Figure 7 c and d show that weak striped patterns exist in the residual results of both the sum of sines and the GAM spline fitting methods, indicating that the high-frequency errors are not completely removed. The MedAD values show that the combination of polynomial and the sum of sines method is 5.1% less accurate than the GAM-based method, which can be observed by the significant negative biases indicated by the arrows in Fig. 7c. Figure 8 further shows the fitting results in the along-track direction. The 8th-order polynomial only matches the long-term trend, and a follow-up experiment revealed that increasing the order of the polynomial still does not help to capture high-frequency signals. As marked by the red and purple arrows in Fig. 8 (corresponding to the regions indicated in Fig. 7), the maximum difference between the sum of sines and the GAM spline fitting results is about 5 m. Because the sinusoidal function is a parametric model whose parameters (amplitude, phase, and frequency) are global constants, there is no difference in shape between the different cycles. In contrast, the GAM spline yields a non-strictly periodic curve by fitting the local relationship between the elevation differences (the response variable) and the along-track coordinates (the predictor variable) over parts of their range. A visual inspection shows that the GAM spline fitting results fit more closely with the local trends in the co-registration residuals, which indicates that the GAM spline fitting method might be a better alternative to the traditional parametric models for residual correction of DEM co-registration results, benefiting from its data-driven nature.

Finally, Table 3 summarizes the residual correction results for the 23 ASTER DEM pairs. The GAM spline fitting method outperforms the polynomial method and the combination of polynomial and the sum of sines method by reducing 4.4% and 2.1% more residuals, respectively. We manually check the residual correction results for all the DEM pairs. A visual inspection shows that the remaining errors for a majority of the data (e.g., pair GrIS-1 in Fig. 6) are almost randomly distributed in the scene. Only a few DEM pairs suffer from minor systematic errors caused by incompletely corrected jitter (e.g., pair GrIS-2 in Fig. 7), where slight biases would be propagated into the glacier thickness change estimates.

**Table 3.** Residual correction results with the 23 DEM pairs of GrIS.

| Method | Average MedAD (m) |
|---|---|
| Polynomial fitting | 5.825 |
| Polynomial and the sum of Sines | 5.686 |
| GAM spline fitting | 5.566 |

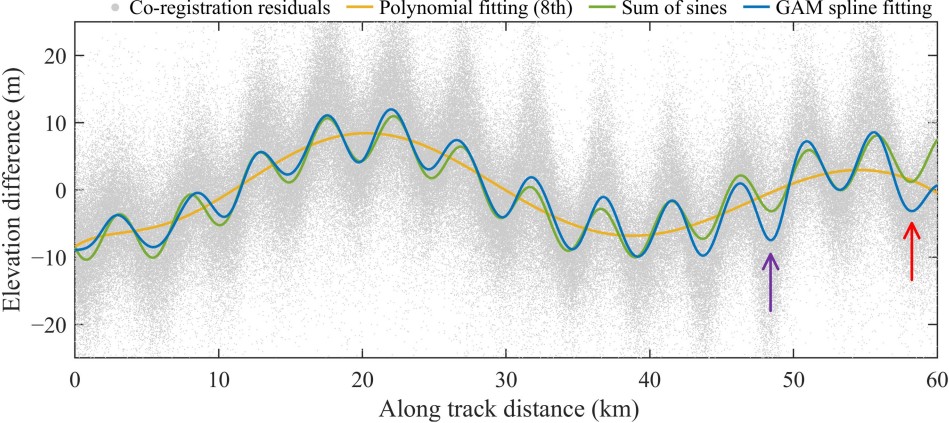

**Figure 7.** Residual correction of DEM pair GrIS-2. **(a)** The DEM co-registration results obtained using the RT method. The residual correction results obtained using polynomial fitting **(b)**, the combination of polynomial and the sum of sines method **(c)**, and GAM spline fitting **(d)**. The black lines mark the GrIS boundaries delineated by Rignot and Mouginot (2012).

**Figure 8.** Co-registration residuals of DEM pair GrIS-2 and the along-track fitting results.

## 4.2. Mountain glacier case study

### 4.2.1. Data processing

The mountain glacier experiments were performed on 22 DEM pairs from the Pamir region of High Mountain Asia (HMA) (Fig. 9), including ASTER DEMs, ZiYuan-3 (ZY-3) DEMs generated from ZiYuan-3 tri-stereo optical scenes (Pan et al., 2013; Liu et al., 2020), and the global Shuttle Radar Topography Mission (SRTM) DEM (Farr et al., 2007) and the Copernicus DEM GLO30 (Airbus, 2020) obtained using the Interferometric Synthetic Aperture Radar (InSAR) technique. Stable regions were extracted from three land cover classes (bare land, artificial surfaces, and cultivated land) in the GlobeLand30 land cover product (Jun et al., 2014; Li et al., 2021).

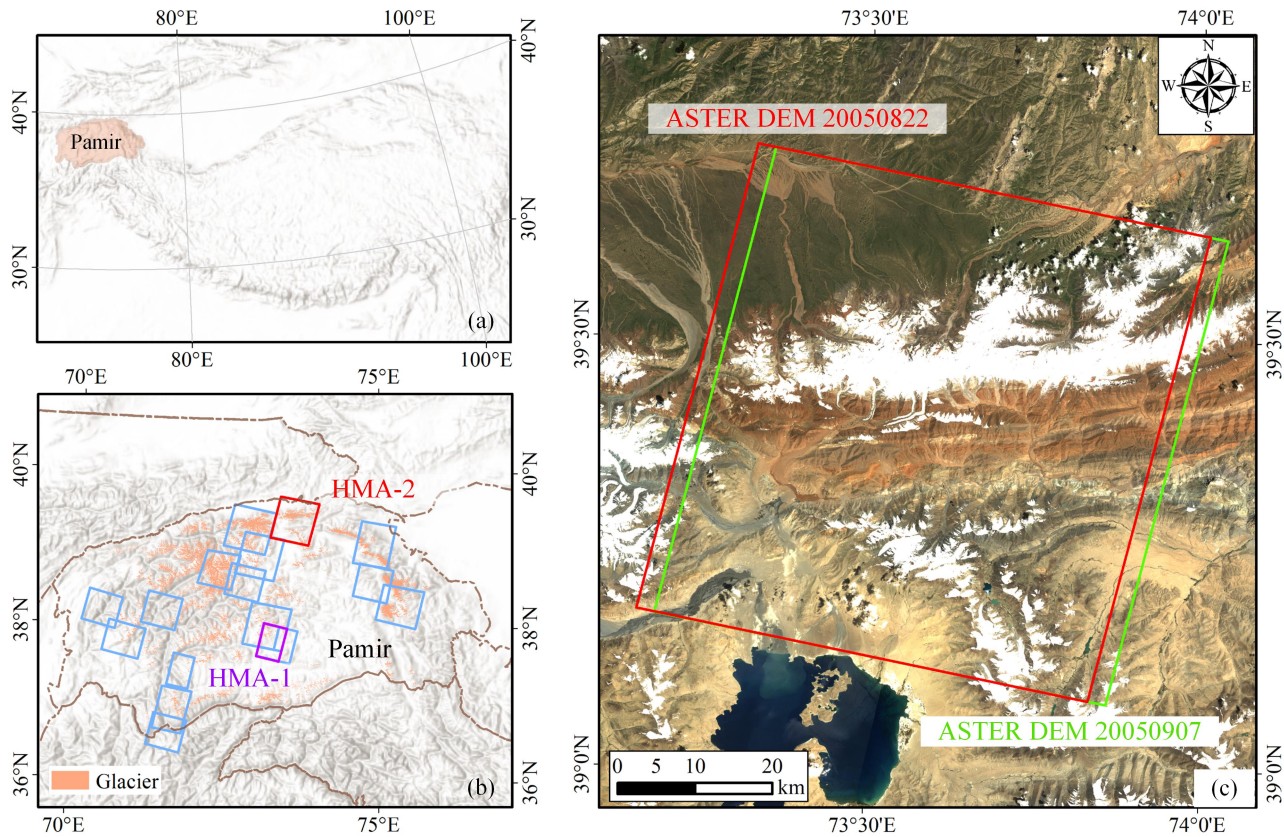

**Figure 9.** The study area located on HMA. (a) and (b) The footprints (blue) of the 22 DEM pairs, where pairs HMA-1 and HMA-2 listed in **Table 4** are highlighted in purple and red, respectively. The glacierized areas are from the RGI 6.0 inventory. (c) The coverage of the two DEM images in HMA-2 (red: ASTER DEM 20050822; green: ASTER DEM 20050907). The background image was acquired by Landsat 5 in 2008.

**Table 4.** Characteristics of the 4 DEM pairs in HMA.

| Pair ID | Data | Roles | Date | Res. (m) | Scene ID |
|---------|------|-------|------|----------|----------|
| HMA-1 | Copernicus DEM | Reference DEM | 2011–2015 | 30 | N37E073, N38E073 |
|  | ZY-3 DEM | Secondary DEM | 8 Oct 2017 | 30 | — |
| HMA-2 | ASTER DEM | Reference DEM | 22 Aug 2005 | 30 | AST14DEM.003:2030590191 |
|  | ASTER DEM | Secondary DEM | 7 Sept 2005 | 30 | AST14DEM.003:2030819798 |

### 4.2.2. DEM co-registration

Like the Ice Sheet case, the simplified and standard versions of the NK method yield similar results for the three test datasets of ZY-3 DEMs, the SRTM DEM and Copernicus DEM in the HMA region (Table 5). The RT method shows better co-registration performance than the three versions of the NK method, with removing 13.7% more errors over the linear version on average.

**Table 5.** Co-registration results obtained with the 22 DEM pairs of HMA.

| Method | Average MedAD (m) |
|--------|-------------------|
| Before co-registration | 15.483 |
| NK standard version | 7.220 |
| NK simplified version | 7.212 |
| NK linear version | 7.220 |
| RT | 6.230 |

Figure 10 depicts an example of the ZY-3 DEM with large attitude errors. From Fig. 10 a to c, it can be seen that the co-registration results of the NK method exhibit significant residuals in the southwest-northeast direction, leading to a false estimate of rapid glacier mass loss in the northern region. In contrast, the RT algorithm can effectively eliminate attitude-induced bias and reduce co-registration errors by 83.3% compared to the NK linear version. More test cases using ZY-3 DEMs are provided in the supplement, and their error-reduction ratios range from 0.7% to 42.3%. A visual comparison between Fig. 10d and Fig. 5d reveals that the co-registration residuals of the ZY-3 DEM are much smaller than those of the ASTER DEM. This may be due to the fact that the ZY-3 raw image has a resolution of 2.5–3.5 m (Zhang et al., 2018) and retains a high signal-to-noise ratio after downsampling to 30 m.

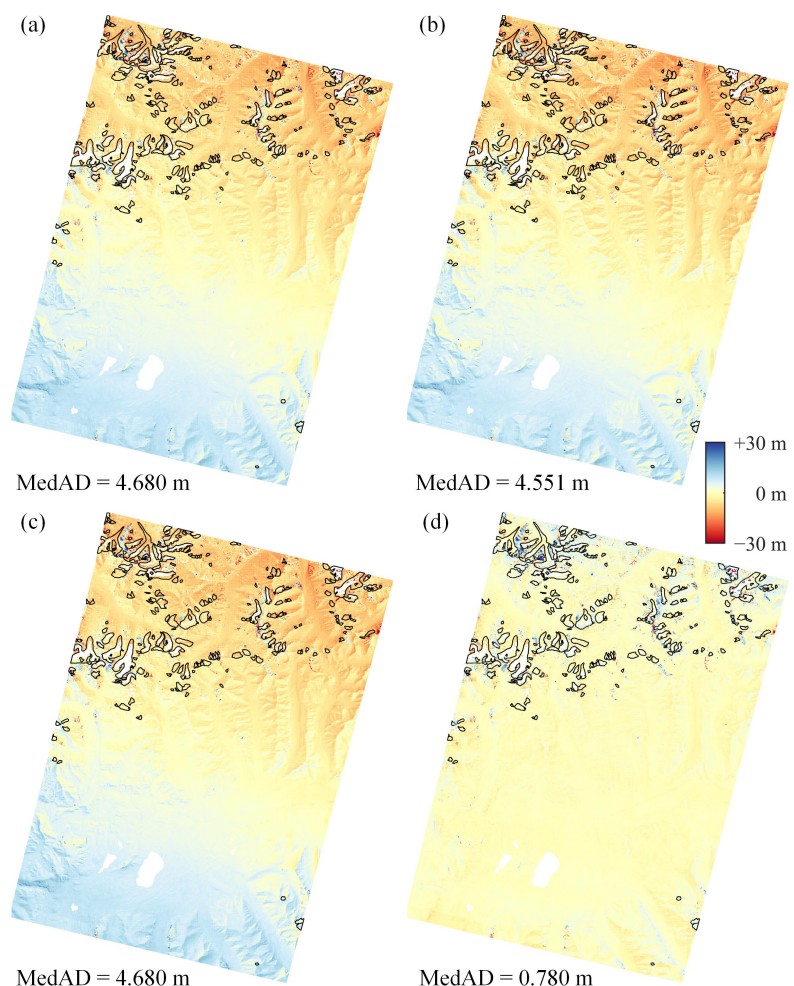

(a)

(b)

MedAD = 4.680 m

MedAD = 4.551 m

+30 m

0 m

−30 m

(c)

(d)

MedAD = 4.680 m

MedAD = 0.780 m

**Figure 10.** Co-registration results of the different methods for DEM pair HMA-1: the standard (a), simplified (b), and linear (c) versions of the NK method, and the RT method (d). The black lines mark the RGI 6.0 glacier outlines.

Figure 11 illustrates the influence of the DEM co-registration method on glacier surface elevation changes estimation. Subplots a, b, and c show the co-registration results and glacier elevation change statistical values for an ASTER DEM pair acquired half a month apart (HMA-2). Since the two DEMs were obtained at a very short time interval, it can be assumed that there is no elevation change. The result of the NK linear version contains obvious rotation-induced errors (Fig. 11a), the large proportion of missing data throughout the center of the image are caused by cloud cover, and the glacier-covered area in the southeast region only covers a small number of invalid pixels. We calculated the glacier elevation change values within the red circle and found a significant negative deviation (−6.396 m). No distinct abnormal trends were found in the results of the RT method, and the mean value of glacier elevation changes is very close to zero.

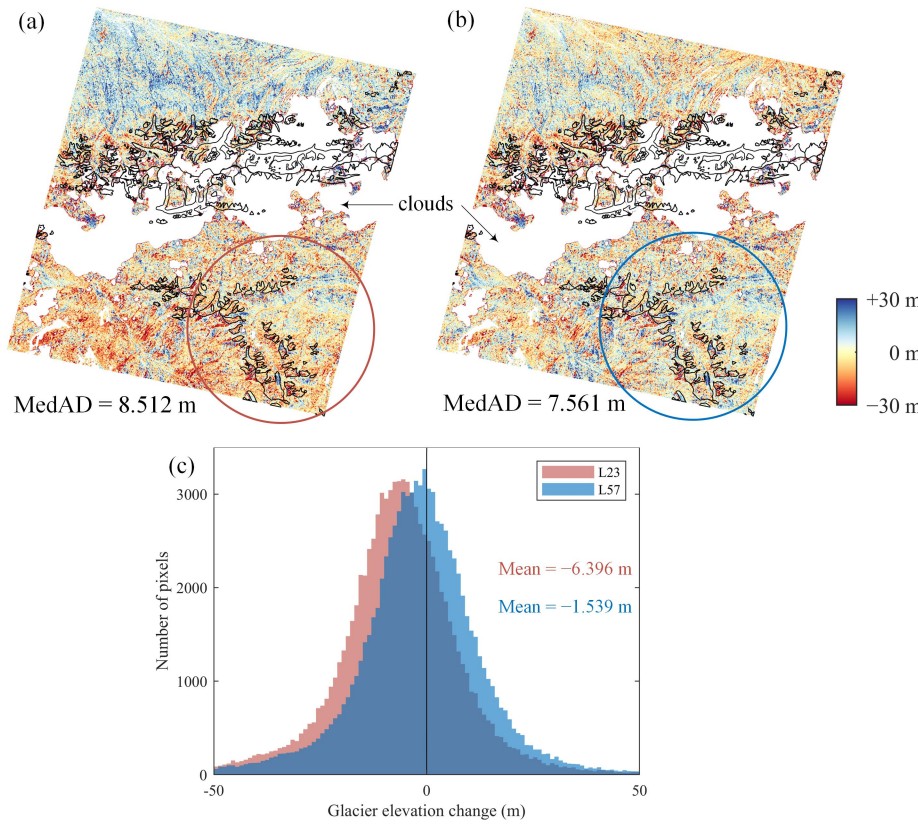

**Figure 11**. Co-registration results of DEM pairs HMA-2 based on linear versions of the NK method (a) and the RT method (b). The black lines mark the RGI 6.0 glacier outlines. (c) The histogram of elevation changes for glaciers within the circle.

We provide DEM co-registration examples in the Supplement for more scenarios in the HMA and New Zealand (NZL),
such as a large number of glaciers, a large amount of vegetation, a high noise level due to rough topography in the DEMs. Since no strong jitter-induced residuals were observed in the co-registration results of these DEM pairs, residual correction experiments were not performed.

## 5. Discussion

The performance of different types of DEM co-registration methods has been intensively investigated by Paul et al. (2015)
and Vacaflor et al. (2022). Their tests showed that the NK method achieved similar or better accuracy compared to many non-analytical methods, such as the grid search method (Berthier et al., 2007), the LS3D method (Gruen and Akca, 2005), and the subwatershed-based method (Li et al., 2017), and therefore the NK method was recommended for practical applications due to the less computational effort (Paul et al., 2015).

This work focuses on the comparison of two analytical algorithms, the NK method and the RT method, which have been
widely used in the cryosphere (Geyman et al., 2022; Hugonnet et al., 2021; Maurer et al., 2019) and photogrammetry (Aguilar

et al., 2012; Zhang et al., 2010) studies, respectively. The characteristics of the two methods are summarized in Table 6, and the theoretical connections and differences between them are discussed in the following.

**Table 6.** Summary of the DEM co-registration methods.

| Method | Regression equation | Explanatory variables | Regression coefficients |
|---|---|---|---|
| NK standard version | (2), nonlinear | $\psi, \theta$ | $a, b, c$ |
| NK simplified version | (6), nonlinear | $\psi$ | $a, b, c'$ |
| NK linear version | (9), linear | $\psi, \theta$ | $\Delta X, \Delta Y, \Delta Z$ |
| RT | (17), linear | $f_X, f_Y, X_C, Y_C, Z_C$ | $\Delta X, \Delta Y, \Delta Z, \gamma, \omega, \varphi, \kappa$ |

1) The form of the regression

The NK method can be expressed as either a nonlinear or linear equation, while the RT method only employs a linear regression model. The disadvantages of nonlinear regression over linear regression are that it works iteratively and often requires more computational resources.

2) The explanatory variables in the regression

The NK method was inspired by the similarity between an elevation difference map and a hillshade, which is predicted based on the terrain slope and aspect. The RT method, on the other hand, employs the terrain gradients (i.e., the partial first derivatives) in the *X* and *Y* directions as the explanatory variables. From Eqs. (15) and (16), it can be seen that the two groups of terrain variables are actually equivalent.

3) Regression coefficients

In the RT method, the misalignment between two DEMs is modeled by a 3-D similarity transformation, including three translation, one scale, and three rotation factors. The NK method considers the spatial shift only, and the regression coefficients can be either cylindrical coordinates ( $a, b, c$ ) or Cartesian coordinates ( $\Delta X, \Delta Y, \Delta Z$ ) of the shift vector.

Based on the above analysis, it can be concluded that the number of regression coefficients is the only significant difference between the NK and RT methods. In other words, the RT method can be viewed as an extension of NK by additionally modeling the scale and rotation errors. The advantage of the original NK method (i.e., the simplified version) is that only one explanatory variable ( $\psi$ ) exists in the regression model, and the shift vector can therefore be calculated by a curve-fitting technique or estimated from a scatter plot, which is easy to adopt for users with a limited knowledge of statistics. However, in terms of precision, the NK method is theoretically inferior to the RT method, because only the shift-induced errors are considered.

The NH method (Noh and Howat, 2014) is another one analytical algorithm that has been previously used in glacial studies, in addition to the NK method. It can be theoretically proven that the NH method is basically equivalent to the RT method (details are available in the supplement), because both of them are derived from a 3-D similarity transformation with considering the errors induced by three translation, one scale, and three rotation factors. The disadvantage of the NH method is that small Euler angles are not approximated, and, accordingly, the regression equation has a very complicated form (see Eqs. S3 and S4 in the supplement), which hinders the replication of the algorithm by other researchers.

To sum up, the three analytical algorithms for the DEM co-registration problem, i.e., NK, NH, and RT, have a strong theoretical relationship, despite being presented in diverse forms in the original literature. Taking into account both the algorithm accuracy and ease of use in practical applications, we recommend to apply the RT method instead of the NK and NH methods in glacial studies.

A residual correction procedure is required after DEM co-registration if satellite attitude jitters and other complex errors
are not be properly removed before the production of DEMs. The residual signals often appear at several frequencies. The numbers are not constant for different DEM sources (Girod et al., 2017; Wang et al., 2016; Ye et al., 2019), and they primarily depend upon sensors and data pre-processing techniques. For the traditional parametric regression methods, the optimize choice of the degree of polynomials (or the number of sinusoidal functions) takes n trials, where n is the highest degree of the polynomial (or the maximum number of sines) allowed. For example, n = 6 suggested by Girod et al. (2017). If the residual
signals in the data do not have a repeated regular shape, the performance of traditional parametric regression methods is limited by their predefined models. The GAM spline fitting method is data-driven and can capture more complex nonlinear patterns in residuals. The spline parameters can be automatically determined by the generalized cross validation or other criterion, and the optimization process requires high computational cost. The GAM spline fitting can be used as an alternative to traditional parametric regression models when the residuals cannot be precisely fitted by high-order polynomials or a sum of sines. It
should be noted that if the noise level of DEM co-registration residuals is very high, the performance of GAM spline fitting method is just slightly better than that of traditional parametric regression methods.

In Section 4, we focused on the accuracy comparison of various algorithms and conducted tests on DEM pairs with good geometric conditions. However, the geometric constraint of stable terrain may be weak for the DEM pairs located at the edge of an Ice Sheet with little stable terrain or covered by heavy clouds. Under such conditions, the reliability issue of algorithms
becomes even more critical.

Figure 12 shows a representative example of one ASTER DEM pair located on the western edge of GrIS, where the stable terrain is geographically distributed in the southwest corner only. The time interval between reference DEM (ASTER DEM 20190725) and secondary DEM (ASTER DEM 20190826) is one month, and therefore the ice surface elevation can be considered unchanged. Although the RT algorithm yields significantly smaller co-registration residuals in the stable region
than the NK method, it is prone to producing larger biases over ice-covered regions. These biases cannot be removed by residual correction procedure, because the residual trend over ice-covered regions is completely different from that over stable regions.

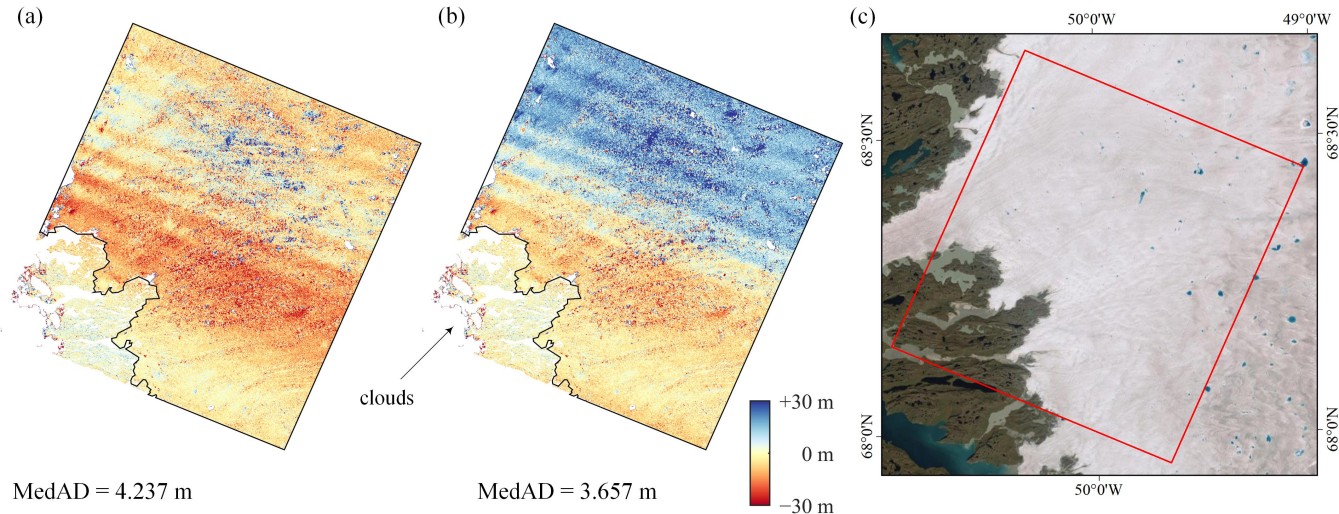

**Figure 12.** Co-registration results of ASTER DEM 20190725 (Scene ID: AST14DEM.003:2344943025) and ASTER DEM 20190826 (Scene ID: AST14DEM.003:2346334895). (a) The NK linear version. (b) The RT method. The black lines mark the ice-sheet boundaries delineated by Rignot and Mouginot (2012). (c) The location of DEM pairs, overlaid on an optical satellite imagery.

The second example is the residual regression of the DEM pair GrIS-19 (in Table S1) by taking the terrain elevation as the explanatory variable. It can be seen from Fig. 13 that the mean elevation of glaciers is much higher than that of bare lands, and a long extrapolation is therefore required. Both the prediction results of the polynomial and spline fitting methods are strongly biased in high altitude regions (> 500 m).

It can be noticed from the above two examples that, in the DEM co-registration and residual correction tasks, unreliable regression results cannot be detected from the elevation differences of stable regions. As a rule of thumb, when a data extrapolation is needed, it is recommended to adopt a conservative strategy by decreasing the degree-of-freedom of the regression model, e.g., dropping some explanatory variables (in DEM co-registration), and reducing the degree of the polynomial (in high-order polynomial regression) or smoothing (in spline regression).

Many researchers pointed out the need for residual correction along terrain altitudes after DEM co-registration. This correction is absolutely necessary for InSAR DEMs to reduce the biases caused by radar penetration (Gardelle et al., 2012; Li et al., 2021), while it is only optional for stereoscopic DEMs (Nuth and Kääb, 2011) because there is no strong physical basis to explain it. Given that obvious elevation-dependent biases were not observed in our experiments (e.g., Fig. 13a), the terrain elevation was not introduced as an explanatory variable in our residual regression model (i.e., its degree-of-freedom is zero), and, accordingly, the problem of data extrapolation along terrain altitudes was circumvented.

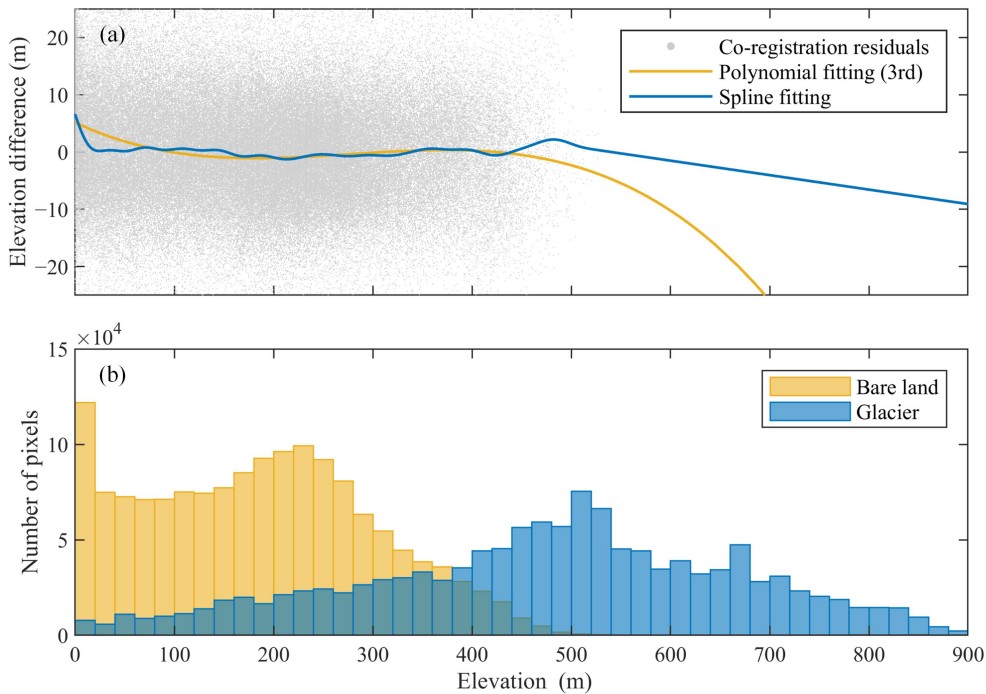

**Figure 13.** Regressions of DEM co-registration residuals against terrain heights. (a) Polynomial and spline fitting results. (b) The histograms of terrain heights for bare land and glacier covered pixels in the overlapping region of DEM pair GrIS-19.

## 6. Conclusion

In this paper, we have made a thorough comparison of the DEM co-registration methods of Nuth and Kääb (2011) and Rosenholm and Torlegard (1988), and proposed a GAM-based method to correct DEM co-registration residuals. The theoretical analysis and experimental results support the following conclusions:

1)  The NK method and the RT method are theoretically compatible with each other. On the one hand, the terrain-related information used by the two methods as explanatory variables in their regressions—slope/aspect and gradient—can be proven to be equivalent through theoretical analysis. On the other hand, even though the methods of NK and RT utilize distinct regression forms, the nonlinear regression equation used by the former can be converted into a linear equation with the same structure as the latter.

2)  Rotation and scale biases should be taken into account in DEM co-registration. The only significant difference between the methods of NK and RT is that the latter models the translation, scale, and rotation-induced biases, while the former only considers the spatial translation. Comparative experiments conducted on multiple DEM pairs showed that the RT method consistently outperformed the NK method in terms of co-registration residuals. Thus, we recommend applying the RT method also in glacial studies.

3) GAM spline fitting can be used as an alternative to traditional parametric regression models in correcting DEM co-registration residuals. ASTER DEMs often suffer from some complex errors with multiple frequencies induced by satellite attitude jitter. Benefiting from its data-driven nature, the GAM spline fitting method can capture the complex nonlinear patterns in DEM co-registration residuals, whereas the performance of the parametric regression methods is limited by their predefined models when the residuals do not have a repeated regular shape.

*Code availability*. Matlab code for DEM co-registration is available at https://github.com/shenapm/DemCoReg and archived at https://zenodo.org/record/8098337 (last access: 22 October 2023).

*Data availability*. ASTER DEMs are freely available at https://search.earthdata.nasa.gov. Landsat 8 images are available at https://glovis.usgs.gov. The Greenland drainage basins delineated by Rignot and Mouginot can be obtained from http://imbie.org/imbie-3/drainage-basins/.

*Supplement*. The supplement related to this article is available online at:

*Author contributions*. XS and TL conceptualized and initiated the study. TL performed the data processing and analyses, prepared the figures and tables, and wrote the draft manuscript. XS contributed to review and improve the manuscript. YH, BL, LJ, and HW assisted to the editing and refining of the manuscript.

*Competing interests*. The authors declare that they have no conflict of interest.

*Financial support*. This research was supported by the National Key R & D Program of China (Nos. 2017YFA0603103 and 2018YFC1406102); the Strategic Priority Research Program of the Chinese Academy of Sciences (No. XDA19070104); the National Natural Science Foundation of China (Nos. 41974009, 42004007 and 42174046); the State Key Laboratory of Geodesy and Earth's Dynamics (Innovation Academy for Precision Measurement Science and Technology, CAS, Nos. S21L6102 and S22L6101); and the Key Research Program of Frontier Sciences, CAS (Nos. QYZDB-SSW-DQC042 and QYZDJ-SSW-DQC027).

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
