# Peer review of "Co-registration and residual correction of digital elevation models: A comparative study"

_The Cryosphere, 2022_

## Author Comment (AC1)

**Response to Reviewer #1 on "Co-registration and residual correction of digital elevation models: A comparative study"**

Comment received: 19 Dec 2022

5 Key:

Reviewer comment (black)

Response (blue)
* * *
Li et al. compared different methods of correcting 3D-shifts and others biases in digital elevation models (DEM) with the

10 ultimate goal to reduce uncertainties in glacier elevation change estimates. They first compared the widely-used Nuth & Kaab (2011) to the less popular Rosenholm and Torlegard (1988) algorithms for DEM coregistration. On top of a simple 3D shift, the latter algorithm also account for any rotation or scale differences between the DEMs. Further, they proposed an improved correction of the structured-biases between the DEMs that have a proper signature in the two directions of image acquisitions (along and across tracks). They go beyond fit by polynomials and sums of sinusoids by proposing a spline-based

15 non parametric model.

I found nothing wrong with this study. However, I am not super convinced that this article, in its current form, fits well in The Cryosphere and its readership. I see two main reasons for that:

1/ DEM differencing is a popular technique to measure glacier changes. However the scope of the present study is really technical with no direct application to glacier changes. The study sites only marginally include glaciers.

20 2/ The added value of the proposed method is modest. I am not convinced that a gain of 0.2 m (5%) in standard deviation of the residuals between DEMs only covering a single (and not very challenging) test site is sufficient to convince the glaciological community to rethink the way they coregister DEMs. The added value of the spline-based correction of along track residuals is higher but would need to be confirmed in different settings.

Overall the paper is well written, the work is performed seriously but I missed some novel results that would make a real

25 impact on the glaciological community.

Thank you for your constructive comments and suggestions.

In the experimental section of the previous manuscript, only the ASTER DEMs on the margins of the Greenland Ice Sheet (GrIS) were used for two main reasons:

1) There is commonly a large proportion of stable terrain in the scene, which is convenient to visually analyze DEM co-

30 registration residuals.

2) Strong jitter-induced errors remain in the DEM co-registration results, which is desirable for the comparison of different residual correction methods.

Our experiments so far have been carried out on more than 200 DEM pairs from ZY-3, ASTER, SRTM, and Copernicus DEMs in GrIS, High Mountain Asia (HMA), and southern Alps (SALP). The comparative results show that the method of Rosenholm and Torlegard has a greater ability to remove DEM misalignments (83.3% maximum) than the method of Nuth and Kääb. Strong jitters can only be observed from the co-registration results of ASTER DEMs in GrIS. The GAM spline fitting method often yields a higher accuracy than the two parametric regression methods (high-order polynomials and the sum of the sinusoidal functions), but the improvement is relatively modest, because the magnitudes of the errors caused by satellite attitude jitters in ASTER DEMs are typically not significantly greater than those of random errors and unmodeled systematic errors. Considering that the residual correction is just a secondary work in our study, the GAM spline fitting method and related experimental results can be completely removed from the manuscript.

Major comments

1/ Do the conclusions apply in other settings? Map of elevation differences are constructed from several ASTER DEMs in western Greenland with a strong proportion of stable terrain. Ice-covered terrain is restricted to the eastern part of the images/DEM. It seems that images are almost cloud free. This site and the cloud-free images are appropriate to design and test the different methods but are not representative of real case scenario. In my experience, further challenges for DEM coregistration comes from: vegetation (changing with time), large fraction of glacier areas vs. stable terrain, gaps or unreliable data in the DEMs due to clouds, the rough topography leading to higher noise level in the DEMs. Authors did not explore these difficulties and thus their results are not representative of more complex and more realistic situations.

Followed by your suggestion, DEM co-registration algorithms have been compared in more complex and challenging scenarios. We present some representative examples below. Since the three versions of the method of Nuth and Kääb always produce similar co-registration results, we only compare the linear version of the method of Nuth and Kääb (L23 in Table 1) and the method of Rosenholm and Torlegard (L57 in Table 1) to make figures clear.

1) Large fraction of glacier areas vs. stable terrain

We select the DEM pair HMA-1 (Table R1) located in the northern Pamirs, excluding the erroneous observation over the bright snow-covered areas (e.g., upstream of Bol and Oktyabrskiy glaciers) due to ASTER saturation issues. The L57 algorithm can effectively remove the residuals in the east-west direction of the L23 results (Fig. R1a) and improve the co-registration accuracy by 17.5%.

**Table R1. Characteristics of DEM pair HMA-1.**

| Pair ID | Data | Date | Res. (m) | Scene ID | MedAD (L23–L57) |
|---------|------|------|----------|----------|-----------------|
| HMA-1 | SRTM | 11–22 Feb 2000 | 30 | N39E072, N39E073, N38E072, N38E073 | 9.768–8.060 |
| | ASTER | 22 Mar 2005 | 30 | AST14DEM.003:2028219582 | (17.5%) |

[Figure]

(a)

(b)

Bol

Oktyabrskiy

+30 m

0 m

−30 m

MedAD = 9.768 m

MedAD = 8.060 m

**Figure R1.** Co-registration results of DEM pair HMA-1: L23 (a) and L57 (b). The black lines mark the glacier boundaries.

65   2)    Gaps or unreliable data in the DEMs due to clouds

As seen in Fig. R2, there are a lot of clouds distributed throughout the center of the ASTER DEM 20050822, and the rotation-induced biases of this image make the glacier in the southeast area of the L23 co-registration results exhibit a positive (Fig. R2a) or negative (Fig. R2b) pattern, which leads to the wrong estimation of the glacier elevation change.

**Table R2. Characteristics of DEM group HMA-2.**

| Group ID | Data | Date | Res. (m) | Scene ID | MedAD (L23–L57) |
|----------|------|------|----------|----------|-----------------|
| HMA-2 | SRTM | 11–22 Feb 2000 | 30 | N39E073 | 8.598–6.366 (26.0%) |
|  | ASTER | 22 Aug 2005 | 30 | AST14DEM.003:2030590191 | 8.512–7.561 (11.2%) |
|  | ASTER | 07 Sept 2005 | 30 | AST14DEM.003:2030819798 | 6.334–5.790 (8.6%) |

70

[Figure]

MedAD = 8.598 m      MedAD = 8.512 m      MedAD = 6.334 m

MedAD = 6.366 m      MedAD = 7.561 m      MedAD = 5.790 m

**Figure R2.** Co-registration results of DEM group HMA-2 based on L23 (top) and L57 (bottom). From left to right: ASTER DEM 20050822 to SRTM DEM, ASTER DEM 20050907 to ASTER DEM 20050822, and ASTER DEM 20050907 to SRTM DEM, respectively.

75   3)   Vegetation

We choose the DEM pair SALP-1 (Table R3) with a large amount of vegetation on the west side (Fig. R3c). After removing the unstable pixels (forest land, water, wetland, and glacier and snow cover), the co-registration results of the L23 method and the L57 method are shown in Fig. R3 a and b. The L57 algorithm can reduce the error in the east-west direction and improve the accuracy by 13.0%.

80

**Table R3. Characteristics of DEM pair SALP-1.**

| Pair ID | Data | Date | Res. (m) | Scene ID | MedAD (L23–L57) |
|---------|------|------|----------|----------|-----------------|
| SALP-1 | SRTM | 11–22 Feb 2000 | 30 | S44E169, S44E170 | 7.898–6.871 (13.0%) |
| | ASTER | 24 Feb 2003 | 30 | AST14DEM.003:2011883607 | |

[Figure]

(a)

MedAD = 7.898 m

(b)

MedAD = 6.871 m

(c)

Stable terrain
Forest land    Wetland
Glacier and snow cover    Water

+30 m

0 m

−30 m

**Figure R3.** Co-registration results of DEM pair SALP-1: L23 (a), L57 (b), and the land cover map using GlobeLand30 product (c).

4)    The rough topography leading to higher noise level in the DEMs

We select the DEM pair HMA-3 (Table R4) with a high noise level in Pamir, and the co-registration accuracy of the L57 method is 8.3% better than that of the L23 method.

**Table R4. Characteristics of DEM pair HMA-3.**

| Pair ID | Data | Date | Res. (m) | Scene ID | MedAD (L23–L57) |
|---------|------|------|----------|----------|-----------------|
| HMA-3 | ASTER | 10 Oct 2017 | 30 | AST14DEM.003:2280543414 | 7.220–6.621 (8.3%) |
|  | ASTER | 26 Oct 2017 | 30 | AST14DEM.003:2281248034 |  |

(a)           (b)

+30 m

0 m

−30 m

MedAD = 7.220 m          MedAD = 6.621 m

**Figure R4.** Co-registration results of DEM pair HMA-3: L23 (a) and L57 (b).

2/ Do the improvements over stable terrain percolate to ice-covered areas? To convince the readers (glaciologists, the readership of TC) of the added values of the proposed methods, authors would need to demonstrate real improvements over glacier terrain.

Such a validation is tricky, I reckon, because glacier elevations are constantly changing. I see two ways for the authors to demonstrate this

(a) apply their methods to DEMs derived from images acquired just a few days apart so that the assumption of no elevation change is almost valid. They would then be in position to coregister and bias correct their DEMs over the stable terrain and then check the improvements on glaciers (where no change should be measured over a few days).

Two DEM pairs have been used for validating the co-registration results over glacier terrain.

The first DEM pair is ZY-3 DEM 20171008 and ASTER DEM 20171010, two days apart (Table R5). The co-registration results of the L23 method show systematic errors in the southwest-northeast direction, resulting in a significantly negative bias (-11.826 m) in the northeast region (within the red circle in Fig. R5a). In contrast, the estimations of glacier elevation changes are much close to zero in the results of the L57 method.

**Table R5. Characteristics of DEM pair HMA-4.**

| Pair ID | Data | Date | Res. (m) | Scene ID | MedAD (L23–L57) |
|---------|------|------|----------|----------|-----------------|
| HMA-4 | ZY-3 | 8 Oct 2017 | 30 | — | 6.126–5.063 (17.4%) |
| | ASTER | 10 Oct 2017 | 30 | AST14DEM.003:2280543414 | |

[Figure]

MedAD = 6.126 m      MedAD = 5.063 m

**Figure R5.** Co-registration results of DEM pair HMA-4: L23 (a), L57 (b), and the histogram of elevation change for glaciers within the circle (c).

[Figure]

**Figure R6.** Co-registration results of ASTER DEM 20050907 and ASTER DEM 20050822 from DEM group HMA-2: L23 (a), L57 (b), and the histogram of elevation change for glaciers within the circle (c).

As shown in Fig. R6, the rotation-induced errors in the results of the L23 method can also be observed in the DEM pair of ASTER DEM 20050907 and ASTER DEM 20050822 (Table R2). We further compared the glacier elevation change estimations of ASTER DEM 20050822 to SRTM DEM and ASTER DEM 20050907 to SRTM DEM. Figure R7 shows that the discrepancy in the co-registration results of the L57 method is smaller than that of L23, with the mean value improving from -5.927 m to -1.669 m.

[Figure]

**Figure R7.** Co-registration results of DEM group HMA-2. ASTER DEM 20050822 to SRTM DEM: L23 (a) and L57 (c); ASTER DEM 20050907 to SRTM DEM: L23 (b) and L57 (d); the histogram of the difference between glacier elevation changes derived from the two DEM pairs (e).

(b) find sites where ASTER DEMs are acquired simultaneously to higher resolution DEMs (for example from the Arctic DEM project) so that a reference elevation change map is available. This second solution is more tricky to identify.

To date, we have not found any ArcticDEM data to meet the needs of validating ASTER DEM co-registration.

3/ The discussion is rather weak. There is a long part about the "extrapolation error" that is mostly unrelated to the rest of the article.

130   In the experiment and discussion sections, we investigated the choice of algorithms in accuracy and reliability, respectively. As shown in Fig. R8, when stable regions are located on one corner of the DEM, the method of Rosenholm and Torlegard is prone to produce incorrect predictions over glacier covered regions. Similar problems exist in all the application scenarios required large extrapolations, e.g., the residual regression example in Fig. 10.

135   **Table R6. Characteristics of DEM pair GrIS-1.**

| Pair ID | Data | Date | Res. (m) | Scene ID | MedAD (L23–L57) |
|---------|------|------|----------|----------|-----------------|
| GrIS-1 | ASTER | 25 July 2019 | 30 | AST14DEM.003:2344943025 | 4.237–3.657 (13.7%) |
| | ASTER | 26 Aug 2019 | 30 | AST14DEM.003:2346334895 | |

[Figure]

**Figure R8.** Co-registration results of DEM pair GrIS-1: L23 (a) and L57 (b).

140   4/ I was also a bit disappointed to see that the techniques are only apply to ASTER DEMs. This also reduce the scope/relevance of the results.

We have conducted the tests on a variety of DEMs, including ZY-3 DEMs, ASTER DEMs, SRTM DEMs, and Copernicus DEMs. Please see the reply to the previous questions for test results and more details.

145   Specific comments

L30. I find it unbalanced that three out of four references on the use of DEMs for glacier elevation change mapping are from Chinese colleagues. Others more seminal papers on the topic could be cited here.

We will replace the latter two references with Gardelle et al. (2013) and Pieczonka et al. (2013).

References:

150   Gardelle, J., Berthier, E., Arnaud, Y., and Kääb, A.: Region-wide glacier mass balances over the Pamir-Karakoram-Himalaya during 1999–2011, The Cryosphere, 7, 1263–1286, https://doi.org/10.5194/tc-7-1263-2013, 2013.

Pieczonka, T., Bolch, T., Junfeng, W., and Shiyin, L.: Heterogeneous mass loss of glaciers in the Aksu-Tarim Catchment (Central Tien Shan) revealed by 1976 KH-9 Hexagon and 2009 SPOT-5 stereo imagery, Remote Sens. Environ., 130, 233–244, https://doi.org/10.1016/j.rse.2012.11.020, 2013.

155

L31. Leprince et al., is about mapping surface displacements (using Cosi-Corr), I am not sure this reference is appropriate for DEM errors. Can authors double check?

We will remove this reference.

160   L50. What are these "scenarios"?

We will change it to "images".

L71. Authors need to explain why they need to revisit the Nuth & Kaab's method and why they present in details these flavours of their method. It is not straightforward for the reader what is the aim here. Also because in the end the results are

165   almost identical...

The following sentences will be added at the beginning of Section 2.

"In this section, we will demonstrate that the method of Nuth and Kääb (2011) and the method of Rosenholm and Torlegard (1988) are theoretically compatible, and the latter can be viewed as an extension of the former by additionally modeling the scale and rotation errors. As the original algorithms in the works of Nuth and Kääb (2011) and Rosenholm

170   and Torlegard (1988) were presented in distinct forms, we will present detailed derivations of the equations used in their algorithms and variants."

Figure 1. I did not really understand the figure because I did not understand what were representing the different letters/segments. Annotation to be clarified.

175   The meanings of $\theta, \psi, a, b, c$ have been annotated in Fig. R9 (highlighted in the green rectangle), $X, Y, Z$ represent the axes along the three dimensions, O,E,G... are points in 3D space, and $dH_{XY}$ and $dH_Z$ are the elevation differences induced by a horizontal shift and a vertical shift, respectively.

Figure R9b (i.e., Fig. 1b in the manuscript) was redrawn from Figure 2 (i.e., Fig. R10) of Nuth and Kääb (2011). It is easy to read but only illustrates the special case when $b = \psi$, where $b$ and $\psi$ are the aspect of the shift vector and the terrain aspect,

180    respectively. Figure R9a presents a 3D illustration of a general case when $b \neq \psi$, which is of importance to interpret the $\cos(b-\psi)$ term in Equation 2 of Nuth and Kääb (2011).

[Figure]

**Figure R9.** Elevation differences induced by DEM shift: **(a)** 3-D view when $b \neq \psi$. **(b)** 2-D view when $b = \psi$.

[Figure]

Figure 2. the terminology "master" and "slave" are not very the best ones for ethical reasons. "Reference" and "secondary" DEMs are better.

We will follow your suggestion.

L204. I do not understand why 23 DEM pairs are first mention and then only 2 DEM pairs are presented in detail in Table 2. Rather include an appendix with the dates and ID of all the DEMs so that the study can be reproduced. Also, as you read in my general comment, a more extensive study using a variety of study sites would be more convincing.

We only displayed the results of 2 DEM pairs to improve the manuscript readability. In the revised version, we will add the information of all DEM pairs in the appendix.

L208. The NDBI index is not often used in the glaciological community so need to be explained a bit more.

As listed in Table R7 (The Table 1 of Nguyen et al., 2021), there are more than 10 bareness indices available in literature. The NDBI index used in this work was adapted from Deng et al. (2015) by replacing the SWIR2 band by SWIR1. Our experimental results show that the revised index performs slightly better than Deng et al.'s version (i.e., the NDSI2 in Table R7) in terms of enhancing bare soil from other land cover features around the periphery of GrIS.

**Table R7. The Table 1 of Nguyen et al. (2021).**

**Table 1.** Bare soil indices derived from Landsat imagery.

| Index | Data | Formula | Case Study | References |
|---|---|---|---|---|
| Bare soil index | Landsat TM, ETM, 8 (OLI) | $BSI = \frac{(SWIR2+R)-(NIR+B)}{(SWIR2+R)+(NIR+B)}$ | The Swiss Plateau, Switzerland | [48] |
| Bare soil index 1 | Landsat TM | $BSI1 = \frac{(SWIR1+R)-(NIR+B)}{(SWIR1+R)+(NIR+B)}$ | Guangdong, China | [34] |
| Bare soil index 2 | Landsat TM | $BSI2 = 100 \times \sqrt{\frac{SWIR2-G}{SWIR2+G}}$ | South Africa | [47] |
| Bare soil index 3 | Landsat TM, ETM | $BSI3 = \frac{(SWIR1+R)-(NIR+B)}{(SWIR1+R)+(NIR+B)} \times 100 + 100$ | Iran | [45] |
| Normalized difference soil index 1 | Landsat TM | $NDSI1 = \frac{SWIR1-NIR}{SWIR1+NIR}$ | – | [46] |
| Normalized difference soil index 2 | Landsat TM | $NDSI2 = \frac{SWIR2-G}{SWIR2+G}$ | Milwaukee and Waukesha, US | [50] |
| Normalized difference bareness index | Landsat TM, ETM | $NDBaI = \frac{SWIR1-TIR}{SWIR1+TIR}$ | Northern coastal China | [52] |
| Bareness Index | Landsat TM | $BI = (R + SWIR1 - NIR)$ | Beijing, China | [44] |
| Enhanced built-Up and bareness index | Landsat ETM | $EBBI = \frac{SWIR1-NIR}{10\sqrt{SWIR1+TIR}}$ | Bali, Indonesia | [31] |
| Modified normalized difference soil index | Landsat 8 (OLI) | $MNDSI = \frac{SWIR2-PAN}{SWIR2+PAN}$ | Dehradun, India | [54] |
| Normalized difference bare land index | Landsat TM, 8 (OLI) | $NBLI = \frac{R-TIR}{R+TIR}$ | Wuhan, China | [53] |
| Dry bare-soil index | Landsat 8 (OLI) | $DBSI = \frac{SWIR1-G}{SWIR1+G} - \frac{NIR-R}{NIR+R}$ | Kurdistan, Iraq | [49] |

R: red wavelength, G: green wavelength, B: blue wavelength, NIR: near-infrared, SWIR1: shortwave infrared band 5 (Landsat TM/ETM) and band 6 (Landsat 8), SWIR2: shortwave infrared band 6 (Landsat TM/ETM) and band 7 (Landsat 8), PAN: panchromatic band 8 (Landsat ETM/8), TIR: thermal infrared band 6 (Landsat TM/ETM) and band 10 (Landsat 8).

205

References:

Deng, Y., Wu, C., Li, M., and Chen, R.: RNDSI: A ratio normalized difference soil index for remote sensing of urban/suburban environments, Int. J. Appl. Earth Obs. Geoinf., 39, 40–48, https://doi.org/10.1016/j.jag.2015.02.010, 2015.

210 Nguyen, C. T., Chidthaisong, A., Kieu Diem, P., and Huo, L.-Z.: A Modified Bare Soil Index to Identify Bare Land Features during Agricultural Fallow-Period in Southeast Asia Using Landsat 8, Land, 10, 231, https://doi.org/10.3390/land10030231, 2021.

L215. I do not understand how this 3-sigma rule is applied to check outliers from the classification. Authors need to
215 elaborate more.

We will edit this sentence as follows:

" …and a three-sigma rule (i.e., more than three times the standard deviation) was employed on the elevation differences between two DEMs to remove erroneous data caused by misclassification of unstable terrain areas."

220 Figure 4 does not really bring much. I think it will be pretty obvious to most readers and can be explained in a few sentences in the text.

We will delete Figure 4 and the relevant descriptions.

L239. Example of why the application of the algorithm to a greater diversity of images is needed.

So far, the simplified and standard versions of the method of Nuth and Kääb have been compared with more than 200 DEM pairs from different sources, and the experimental results show that their performance is always close to each other. However, we still cannot reach a more definitive conclusion, because it lacks theoretical grounding.

L302. English is not really correct I think. Check

We will remove this sentence.

L320. This statement (and the rest of the paragraph) about extrapolation error and elevation error as a function of altitude comes a bit out of nowhere. Why discussing extrapolation when this was not mentioned before.

Please see the response to the third question of Major comments.

L328. This is exactly what the revised study should do: include cases where bare terrain is rare and see how the different methods compare.

There are two cases where bare terrain is rare:

1)  Stable regions are fairly evenly distributed throughout the image

Compared to the L23 method, L57 usually yields a higher co-registration accuracy by additionally modeling the scale and rotation errors (refer to Fig. R3).

2)  Stable regions are located on one side or even on one corner of the image

Due to data extrapolation, both the L23 and L57 methods are prone to yield unreliable co-registration results (refer to Fig. R8).

L349. I found the improvements rather modest. "outperformed" is a bit overselling.

The additional test results for HMA and SALP showed that the L57 algorithm could improve the co-registration accuracy by up to 83.3% more than the L23 method. Here we illustrate two examples in HMA (Figs. R11 and R12), where the DEM information used is shown in Table R8.

**Table R8. Characteristics of the two DEM pairs.**

| Pair ID | Data | Date | Res. (m) | MedAD (L23–L57) |
|---------|------|------|----------|-----------------|
| HMA-5 | Copernicus | 2011 to 2014 | 30 | 4.680–0.780 (83.3%) |
|  | ZY-3 | 8 Oct 2017 | 30 | |
| HMA-6 | Copernicus | 2011 to 2014 | 30 | 2.476–1.429 (42.3%) |
|  | ZY-3 | 2 Dec 2016 | 30 | |

[Figure]

MedAD = 4.680 m          MedAD = 0.780 m

+30 m
0 m
−30 m

**Figure R11.** Co-registration results of DEM pair HMA-5: L23 (a) and L57 (b).

[Figure]

MedAD = 2.476 m          MedAD = 1.429 m

+30 m
0 m
−30 m

255    **Figure R12.** Co-registration results of DEM pair HMA-6: L23 (a) and L57 (b).

---

## Author Comment (AC2)

**Response to Reviewer #2 on "Co-registration and residual correction of digital elevation models: A comparative study"**

Comment received: 11 Jan 2023

5  Key:

Reviewer comment (black)

Response (blue)
* * *
This paper deals with the co-registration of DEMs for determining surface elevation changes by DEM differencing. The
10  differences between DEMs are affected by random measuring errors and various systematic errors due to imperfections of the
sensors. The success of the simple differencing method depends on how well the systematic errors can be determined and be
removed. The paper begins with comparing 3 variants of the Nuth and Kääb method with the lessor known method proposed
by Rosenholm and Torlegard. The authors then introduce a non-parametric residual correction model and present results from
a few experiments with ASTER DEMs of Western Greenland.

15  Thank you for your constructive comments and suggestions.

Major Comments

I agree with the authors that the Nuth and Kääb method is predominently used for co-registering ASTER DEMs by the
cryospheric research community. The method has been improved over the years, particularly with handling systematic errors
20  of ASTER DEMs (see reference Luc Girod et al., 2017). If one wants to correct Aster DEMs, as the authors do, then I think
one should start the process on the level of 2017 (see reference above) and not on the original level of 2011. The main reason
is that in the 2017 version a new DEM is computed (MMASTER) with superior image matching that increases the reliability
of the DEMs.

The research objects of our work are the DEM co-registration and residual correction methods utilized by cryospheric research
25  for DEM differencing. In the previous manuscript, we chose the ASTER DEMs as a test dataset, which existed obvious
complex systematic errors induced by satellite altitude jitter, enabling us to simultaneously design co-registration and residual
correction methods. In the additional experiment, we tested DEMs widely used by cryospheric research, including ZY-3 DEMs,
SRTM DEMs, and Copernicus DEMs (please see the response to Reviewer #1 for specific experimental results).

The research of Girod et al. (2017) includes two points: the process from a single stereo pair (one ASTER L1A scene) to an
30  ASTER DEM and the correction of DEMs differences (dDEMs). Since we are not involved in the ASTER L1A data processing,
the first point is not compared. For the dDEMs process, Girod et al. (2017) first adopted the DEM co-registration method
described by Nuth and Kääb (2011), then proposed a parametric regression model (the sum of the sinusoidal functions) to
correct the jitter-induced bias, which we have compared in the previous manuscript (Sect. 3.1. Parametric regression).

Another comment is related to the 'master/slave' concept to co-register DEMs. As is apparent from Table 2, the authors use as a master another ASTER DEM. That makes all computations relative to the master (which is essentially affected by the same systematic sensor errors as the slave) and thus precluding comparisons to an accepted ground reference system. In the area of the test site in Greenland are alternative sources that would be much better suited for serving as a master DEM (e.g. ICESat-2, World View DEMs, ATM airborne laser altimetry).

We agree that refined ASTER DEMs, ICESat-2, and ArcticDEM would be better alternatives for the accurate estimation of glacier changes. However, it should be noted that the main goal of this work is not to access the accuracy of new DEM datasets, but to compare different DEM co-registration algorithms. So far, our experiments have been carried out on four different DEM sources, including ASTER, ZY-3, SRTM, and Copernicus DEMs. The georeferencing errors in these DEMs are commonly larger than those in up-to-date data, but much smaller than those in the historical data before 2000, which is adequate for the comparative analysis of different DEM co-registration methods.

The research results presented in this paper include a comparison between the methods proposed by Nuth/Kääb and Rosenholm/Torlegard. The results of these comparisons can be found in Table 3. The numbers confirm what other researchers have found. The question I have is the definition of AverageMed which is used throughout the paper. How does it compare with more traditional statistical error measures such as mean, median, standard deviation?

The mean and median are measures of location, while the standard deviation is a measure of scale (Rousseeuw and Hubert, 2018). There are two versions of MedAD (the median of all absolute deviations, also abbreviated as MAD) used in literature, i.e., MedAD around the median (Mcmillan et al., 2019) and around zero (Shen et al., 2021).

The MedAD around the median is a measure of scale and can be seen as a robust version of the standard deviation. It is calculated as follows:

$$\text{MedAD} = 1.4826 * \underset{i=1,\dots,n}{\text{median}}\left(\left|x_i - \underset{j=1,\dots,n}{\text{median}}(x_j)\right|\right) \tag{1}$$

where $x = H_{\text{Master}} - H_{\text{Slave}}$ in our manuscript. The constant 1.4826 is a correction factor which makes the MedAD consistent with the standard deviation at Gaussian distributions (Rousseeuw and Hubert, 2018).

In our manuscript, the MedAD is calculated around zero, and the constant 1.4826 is omitted:

$$\text{MedAD} = \underset{i=1,\dots,n}{\text{median}}\left(\left|x_i\right|\right) \tag{2}$$

This form of MedAD is a combined measure of location and scale, and it can be used as a robust alternative to the Root-Mean-Square Deviation (RMSD).

As shown in the last four rows of Table R1, the ratio of the standard deviation (Std) to the MedAD around zero is very close to 1.4826 in our experiments, because the distribution of DEM co-registration residuals is often nearly Gaussian with a zero mean.

**Table R1.** Co-registration results obtained with the 23 DEM pairs.

| Method | ID | Average Median (m) | Average MedAD (m) | Average Mean (m) | Average Std (m) |
|---|---|---|---|---|---|
| Before co-registration | — | −3.391 | 12.043 | −3.504 | 12.604 |
| Nuth and Kääb standard version | N23 | 0.045 | 7.170 | −0.014 | 10.910 |
| Nuth and Kääb simplified version | N13 | 0.036 | 7.163 | −0.017 | 10.913 |
| Nuth and Kääb linear version | L23 | 0.045 | 7.170 | −0.014 | 10.910 |
| Rosenholm and Torlegard | L57 | 0.005 | 6.839 | 0.002 | 10.484 |

References:

McMillan, M., Muir, A., Shepherd, A., Escola, R., Roca, M., Aublanc, J., Thibaut, P., Restano, M., Ambrozio, A., and Benveniste, J.: Sentinel-3 Delay-Doppler altimetry over Antarctica, The Cryosphere, 13, 709–722, https://doi.org/10.5194/tc-13-709-2019, 2019.

Rousseeuw, P. J. and Hubert, M.: Anomaly detection by robust statistics, WIREs Data Mining and Knowledge Discovery, 8, e1236, https://doi.org/10.1002/widm.1236, 2018.

Shen, X., Ke, C.-Q., Yu, X., Cai, Y., and Fan, Y.: Evaluation of Ice, Cloud, And Land Elevation Satellite-2 (ICESat-2) land ice surface heights using Airborne Topographic Mapper (ATM) data in Antarctica, Int. J. Remote Sens., 42, 2556–2573, https://doi.org/10.1080/01431161.2020.1856962, 2021.

Another conclusion the authors make is that GAM spline fitting can be used to reduce complex systematic errors that are still present after geo-referencing. These research results are OK but limited to a specific sensor (ASTER, 25 years in space, outdated technology, complex suite of systematic errors that change in time). Moreover, since GAM spline fitting seems to play an important role in this paper I would strongly suggest to cover it in more detail and provide readers with explanations why you choose it.

Thank you for your suggestion. More technical details of the GAM spline fitting algorithm will be included in the revised manuscript.

Our experiments so far have been conducted on more than 200 DEM pairs from four different sources including ASTER, ZY-3, SRTM, and Copernicus DEMs, located in the Greenland Ice Sheet (GrIS), High Mountain Asia (HMA), and southern Alps (SALP). The experimental results show that strong jitters can only be observed in the ASTER DEMs of GrIS.

Though the paper is written well I have doubts that it is suitable for publishing in this journal in its current form. The methodology presented in this paper should be made more relevant to cryospheric research or might be better suited for a journal that is more focused on new methods and algorithms.

The paper of Nuth and Kääb (2011) was originally published in "The Cryosphere". Their algorithm is currently the most commonly used DEM co-registration method in glacial studies, but it has not been widely adopted in other geoscience

applications. Our work aims to present a deep investigation on the algorithm of Nuth and Kääb, and, therefore, we submitted

95    our manuscript to the same journal, "The Cryosphere".

As suggested by you and Reviewer #1, we will provide more experiments to investigate the choice of DEM co-registration algorithms on glacier change estimation. For more details, please see our replies to Reviewer #1.

Reference:

100    Nuth, C. and Kääb, A.: Co-registration and bias corrections of satellite elevation data sets for quantifying glacier thickness change, The Cryosphere, 5, 271–290, https://doi.org/10.5194/tc-5-271-2011, 2011.

---

## Author Response (AR2)

**Response to Reviewer #1 on "Co-registration and residual correction of digital elevation models: A comparative study"**

Comment received: 31 May 2023

5 Key:

Reviewer comment (black)

Response (blue)
* * *
They are cases where the authors do the minimum to reply to reviews and this is frustrating for the reviewers. And there are
10 others cases where they take into account all reviewer's comments, reply to them in details and do even more than what was requested. We fall here in the latter situation and I would like to command the authors for that. It makes the life of a referee much easier. By processing numerous DEMs from different sources and in different contexts, authors managed to demonstrate that the alternative DEM coregistration method by Rosenholm and Torlegard should be now seriously considered by the glaciological community.

15     Thank you for your detailed review and constructive comments.

My only major criticism concerns the discussion which is very short, weak and include an analysis of the bias with altitude that is not well connected to the rest of the article. Maybe such a result could fit into the discussion but currently it lacks a clear connection to the rest of the paper. Authors need to explain that such a correction is sometime applied in the literature
20 after the two others corrections (refs needed) but without strong physical basis. Then they can discuss to what extent this is a difficult correction to apply as in most cases the calibration sample (stable terrain) is so different from the target sample (glaciers) in term of altitude.

      Some changes have been made according to your suggestions.

      Reliability is the common topic for all the contents in the discussion section, while the experimental section focuses on
25 accuracy only.

Authors followed my suggestion to examine DEMs separated by just a few days. They describe the remaining bias on glaciers (histogram in Figure S2) but there is NO reason to compute it on a specific area of the image only. Residuals on the entire image need to be computed. The histogram is useful but authors could also provide some metrics like the mean and the
30 standard deviation for elevation changes on glaciers for both methods. The mean is important here because this is ultimately what we want to measure (the glacier mean elevation changes).

      Figure R2a shows the elevation-change histogram of the glaciers from the entire image in Figure R1. The results of the
      L23 method are sufficiently negatively biased. The reason for this is that almost all of the glaciers locate on the north side (cf.

Figure R1). However, if glacier-covered pixels were uniformly distributed on the image, the mean of elevation changes on
35 the glaciers would be nearly zero, like the statistical result on stable regions (Figure R2b). Though the performance
difference between L23 and L57 methods can also be observed from the standard deviations, similar information has been
previously provided by MedAD values in Figure R1.

Figure R2c demonstrates that how large biases can be caused by uncorrected attitude errors, which is why only a
specific area of the image was used for the calculation in our manuscript. It can be seen that a bias of approximately 10 m
40 exists in a local area, while the MedAD difference on the entire image is just about 1 m (6.126 m vs. 5.063 m in Figure R1).

[Figure]

(a)  (b)

MedAD = 6.126 m          MedAD = 5.063 m

+30 m

0 m

−30 m

Figure R1. Co-registration results of DEM pair HMA-6: L23 (a), L57 (b)

[Figure]

Figure R2. Histograms of elevation differences in Figure R1. Top: the entire image. Bottom: the specific area within the circle. Left: Glacier-covered regions. Right: Stable regions.

Authors need to tell how, practically, readers could access their tools to apply the Rosenholm and Torlegard coregistration methods. Do they have an implementation of the algorithm open to all? In which language?

The codes are written in MATLAB, currently available at https://github.com/shenapm/DemCoReg, and archived at https://zenodo.org/record/8098337.

Specific comments

Line numbers refers to the track change version of the MS uploaded by the authors *-ATC2.pdf

Abstract : The 83.3 % improvement is true but this an outlier. It can be quoted in the main text but it would be important to quote the mean improvement in the abstract. To avoid overselling.

This sentence has been revised to "an average of 4.6% and 13.7% for the test datasets from Greenland Ice Sheet and High Mountain Asia, respectively".

Figure 1. I had no problem visualizing and understanding the NK-2011 figure but struggle here to visualize this one even in 2D.

The 2D graph (Fig. 1b) of our manuscript is compatible with the Fig. 2 of NK-2011 (Figure R3b vs. Figure R4a). In our drawing, the elevation difference is greatly exaggerated, which aims to make it readily interpreted in a 3D graph (Figure R3a).

For ease of understanding, Figure R3b can be divided into two parts: a DEM is first shifted horizontally (Figure R5a) and then shifted vertically (Figure R5b). The induced elevation difference is given by

$$dH = dH_{XY} + dH_Z$$
$$= a \cdot \tan(\theta) + c \tag{R1}$$

The above equation is valid only for the special case when $b = \psi$, while the correct equation for the general case (i.e., the Equation 2 of NK-2011 and the Eq. 2 in our manuscript) should be

$$dH = a \cdot \cos(b - \psi) \tan(\theta) + c \tag{R2}$$

The multiplier term $\cos(b - \psi)$ in Eq. (R2) can only be illustrated by a 3-D graph (i.e., Figure R3a). In the general case, the shift vector (red arrow) and the DEM gradient vector (blue arrow) located at different vertical planes (OE'G' vs. OEF). In other words, their aspect angles are different with each other, i.e., $b \neq \psi$. It can be seen from Figure R3a that the horizontal shift vector OE' is composed of EE' and OE. On the one hand, EE' does not cause any elevation change, because it is perpendicular to the vertical plane OEF defined by the gradient vector and the terrain aspect direction. On the other hand, the elevation change induced by the vector OE is given by

$$dH_{XY} = EF = OE \cdot \tan(\theta) = OE' \cdot \cos(b - \psi) \tan(\theta) = a \cdot \cos(b - \psi) \tan(\theta) \tag{R3}$$

[Figure]

Figure R3. The Fig. 1 of our manuscript. Elevation differences induced by DEM shift. (a) 3-D view when $b \neq \psi$. (b) 2-D view when $b = \psi$.

[Figure]

[Figure]

Figure R4. The Fig. 2 of Nuth and Kääb (2011). The original version (a) and a revised version with some auxiliary information (b).

[Figure]

85    Figure R5. Decomposition of Figure R3b. Elevation differences induced by a horizontal shift (a) and a vertical shift (b) of a DEM.

Table 1. It would be easier for the reader to label Nuth and Kaab using NK and Rosenholm and Torlegard using RT all along the article.

    Modified as suggested.

90

L183. Should Leprince et al., 2007 be retained here and elsewhere? They do not deal with DEM (but velocity fields). This ref could be quoted of course but authors need to explain why it is relevant to the present study then.

Thanks for pointing it out. This reference has been removed.

95 L236. The classical reference for the NMAD is Holhe and Hohle (cited a lot in the crysophere community, Höhle, J. and Höhle, M.: Accuracy assessment of digital elevation models by means of robust statistical methods, ISPRS Journal of Photogrammetry and Remote Sensing, 64, 398–406, https://doi.org/10.1016/j.isprsjprs.2009.02.003, 2009) that defined it using the 1.4826 scaling factor. Maybe make it 100% clear that this factor is not used.

There are several variants of the MedAD (a.k.a. MAD) in literature, e.g.,

100 1) $1.4826 \cdot \underset{i=1,\ldots,n}{\mathrm{median}} \left( \left| x_i - \underset{j=1,\ldots,n}{\mathrm{median}}(x_j) \right| \right)$

2) $\underset{i=1,\ldots,n}{\mathrm{median}} \left( \left| x_i - \underset{j=1,\ldots,n}{\mathrm{median}}(x_j) \right| \right)$

3) $\underset{i=1,\ldots,n}{\mathrm{median}} \left( \left| x_i \right| \right)$

In the first two variants, the MedAD is calculated around the median. It is a measure of scale and can be seen as a robust version of the standard deviation. The constant 1.4826 is a correction factor which makes the MedAD consistent with
105 the standard deviation at Gaussian distributions.

The last variant was used in our experiments. It is calculated around zero and can be used as a robust alternative to the Root-Mean-Square Deviation (RMSD). This version of MedAD is a combined measure of location and scale, and, accordingly, the 1.4826 factor should not be introduced.

110 L274. I do not think accuracy is the appropriate term here. The authors measured the improvement as a decrease in the dispersion of the residuals which is a proxy for the precision.

This sentence has been revised to "with removing 11.8% more errors".

L356. Fig 5d. A typo here? Authors also need here to tell (L355) that 83% improvement is an extreme case. Otherwise, it
115 oversells the result of the study.
L358. I do not think image resolution has something to do here. It is a more modern satellite platform with better known orbital parameters. It is likely why the initial offsets are smaller than with ASTER.

Some statistical results of the other ZY-3 DEMs in the supplement have been added to the main text according to your suggestion.

Many studies have shown that ZY-3 (a.k.a. ZiYuan-3) satellite suffers from large attitude errors (Cao et al., 2017; Shen et al., 2017; Ye et al., 2019). In this study, more attitude-induced residuals can be overserved in ZY-3 DEMs, e.g., 4.680 m (Fig. 10c) → 0.780 m (Fig. 10d), when comparing to those in ASTER DEMs, e.g., 6.963 m (Fig. 5c) → 6.140 m (Fig. 5d).

When the RT method is used, the co-registration residuals of ZY-3 DEMs are much smaller than those of ASTER DEMs, e.g., 0.780 m (Fig. 10d) vs. 6.140 m (Fig. 5d). The reason for it may be a higher resolution of ZY-3 raw images, rather than the attitude measurement accuracy.

Reference:

Cao, J. S., J. H. Fu, X. X. Yuan and J. Y. Gong (2017). "Nonlinear bias compensation of ZiYuan-3 satellite imagery with cubic splines." ISPRS Journal of Photogrammetry and Remote Sensing 133: 174-185.

Shen, X., B. Liu and Qing-Quan (2017). "Correcting bias in the rational polynomial coefficients of satellite imagery using thin-plate smoothing splines." ISPRS Journal of Photogrammetry and Remote Sensing 125: 125-131.

Ye, Z., Y. S. Xu, X. H. Tong, S. Z. Zheng, H. Zhang, H. Xie and U. Stilla (2019). "Estimation and analysis of along-track attitude jitter of ZiYuan-3 satellite based on relative residuals of tri-band multispectral imagery." ISPRS Journal of Photogrammetry and Remote Sensing 158: 188-200.

Figure 11. For the sake of simplicity I would simply show panel b, e, g. It will make the life of your readers much easier. In fact I do not really see the point of introducing the coregistration to SRTM here. But authors need to compute the glacier elevation changes statistics (mean and standard deviation) for the entire set of glaciers in the scene and write these numbers close to the histograms.

This figure has been modified based on your suggestion. The reason that only a corner area of the image was used for the calculation has been explained in the reply to your earlier comment.

[Figure]

Figure R6. Co-registration results of DEM pairs HMA-2 based on linear versions of the NK method (a) and the RT method (b). (c) The histogram of elevation change for glaciers within the circle.

145

Figure 12. Authors need to add a third panel to show how one of the image looks like so that the reader can understand what is bare terrain and where the ice sheet is.

Modified as suggested.

[Figure]

(a)

(b)

(c)

MedAD = 4.237 m

MedAD = 3.657 m

clouds

+30 m

0 m

−30 m

Figure R7. Co-registration results of ASTER DEM 20190725 (Scene ID: AST14DEM.003:2344943025) and ASTER DEM 20190826 (Scene ID: AST14DEM.003:2346334895). (a) The NK linear version. (b) The RT method. (c) The location of DEM pairs, overlaid on an optical satellite imagery.

L424. This does not fit well with the rest of the article as it was clearly explain that correction of DEM with altitude would not be performed. I agree with the results but I think they do not fit in well here. So basically, after removal of this part, the discussion would be limited to just some new results to show the limitation of the coregistration. This makes a very thin discussion. I think the analysis of DEMs with reduced time difference to confirm the relevance of the RT methods for glacier change study could nicely fit in the discussion.

In Sections 4 and 5, we focus on two different topics, i.e., accuracy and reliability, respectively. There was no discussion section in our initial manuscript. The editor suggested that it might be better for us to add a discussion section, so we decided to include some reliability-related stuff, considering that almost all the contents related to algorithm accuracy have been already discussed in the experimental section.

The reliability issue caused by a lack of sufficient geometric constraints occurs in various regression problems, including DEM co-registration as well as residual correction along terrain altitudes and planimetric coordinates. We agree with the reviewer that these discussions are not strongly related to other parts of the paper. If the reviewer still thinks that this part is too weak, we could remove the whole section and change the title of Section 4 to "Experimental results and discussions".

Supplement. Define abbreviation (case study sites) to make the supplement self consistent.

Modified as suggested.

---

## Author Response (AR3)

**Response to Editor on "Co-registration and residual correction of digital elevation models: A comparative study"**

Comment received: 26 July 2023

5  Key:

Editor comment (black)

Response (blue)
* * *
Thank you for the further improvement. Most concerns raise were adequately addressed. However, the major criticism about

10  the (too) short discussion was only weakly addressed. The related response "Some changes have been made according to your suggestion" is also not convincing. The authors should write more specifically what they improved.

In general, in this section the authors should provide a more in depth discussion about the pros and cons of the NK and RT methods, the non parametric approach to remove complex systematic errors along with the current literature. They also might consider to move or reconsider some of the parts of 2.3 and discuss it along with the own findings of the comparisons.

15  Thank you for your constructive comments. All contents in Section 2.3 have been incorporated into Section 5, and the pros and cons of all methods have been clearly stated in the revised manuscript and listed by the following table.

| Task | Method | Pros | Cons |
|---|---|---|---|
| DEM co-registration | NK | Easy to adopt for users with a limited knowledge of statistics | Scale- and rotation- induced errors are not considered |
| | RT | High precision | None |
| Residual correction | Parametric | Easy to understand | Limited by the predefined model |
| | Non parametric | High precision | High computational cost |

20  In addition, I have few technical corrections. Note that the line numbers refer to the track change version.

General comment: Include the source of the glacier outlines in the figure captions where shown.

Changes have been made according to your suggestions.

The GrIS boundaries are delineated by Rignot and Mouginot, available at the IMBIE website. The glacier outlines are obtained from the RGI 6.0 (for the peripheral glaciers in Greenland, the connectivity level 2 data are excluded).

25

L21: from "the Greenland Ice Sheet"…

L22: the SRTM and Copernicus DEM (include "the" and singular)

L32: I suggest to include the global study by Hugonnet et al. (2021), Nature

L70f: The sentences "The rest of this paper is organized as follows. Sections 2 and 3 introduce and analyze the main co-registration and residual correction methods. Section 4 provides the experimental results, and Sect. 5 concludes the paper." Are not needed. Remove.

L80 Use the abbreviations NK and RT also

L 152: The heading "Discussion" is misleading. You basically compare the two methods. Adjust the heading accordingly

323: Again: SRTM and Copernicus DEM (singular)

Figure 9: The satellite image is blurred. Make sure it is of high resolution for the final version.

L. 377: Remove "etc."

L 381: Mention the specific section number.

L382: Substitute "In real-world" with "However" and omit "very"

At the edge of "an Ice Sheet with little stable terrain"

Modified as suggested. Thanks for your detailed review.

---

## Author Response (AR4)

**Response to Editor on "Co-registration and residual correction of digital elevation models: A comparative study"**

Comment received: 13 Sep 2023

5  Key:

Editor comment (black)

Response (blue)

First of all I appologise for the long time needed to get back to you. This was due to an illness which lasted several
10  weeks and prevented me from working.

The manuscript has further improvedas as did the discussion section. However, I am still missing a discussion "along with the current literature" which I specifically demanded. Moreover, some statements is the discussions are to vague. Some parts of the discussion are also relevant for the conclusions.

The discussion section has been revised to include relevant literature.

15  Our work focuses on the analytical (i.e., the terrain information based) methods only, given that previous studies (Paul et al., 2015; Vacaflor et al., 2022) have already shown that the NK method is more recommended for glacier research than other DEM co-registration methods, such as the grid search method, the LS3D method, and the subwatershed-based method.

The RT method has not been investigated by the cryosphere community, and there is no previous literature to discuss the difference between the RT and NK methods.

20  In addition to the NK method, the NH method (Noh and Howat, 2014) is the only one analytical algorithm that has been previously used in glacial studies. The theoretical connections between the NH and RT methods have been discussed in the revised manuscript. The NH method has not been widely adopted by other researches, possibly because its equations are quite long and tedious (see the Equation S4 in the supplement). We recommend the RT method rather than the NH method, because the two methods are basically equivalent to each other (i.e., both based on a 3-D similarity transformation, rather
25  than the shift used in NK method), while the former is much easier to implement in programming.

**References:**

Paul, F., Bolch, T., Kääb, A., Nagler, T., Nuth, C., Scharrer, K., Shepherd, A., Strozzi, T., Ticconi, F., Bhambri, R., Berthier, E., Bevan, S., Gourmelen, N., Heid, T., Jeong, S., Kunz, M., Lauknes, T. R., Luckman, A., Merryman Boncori, J. P.,
30  Moholdt, G., Muir, A., Neelmeijer, J., Rankl, M., VanLooy, J., and Van Niel, T.: The glaciers climate change initiative: Methods for creating glacier area, elevation change and velocity products, Remote Sens. Environ., 162, 408–426, https://doi.org/10.1016/j.rse.2013.07.043, 2015.

Noh, M. and Howat, I. M.: Automated Coregistration of Repeat Digital Elevation Models for Surface Elevation Change Measurement Using Geometric Constraints, IEEE Trans. Geosci. Remote Sensing, 52, 2247–2260, https://doi.org/10.1109/TGRS.2013.2258928, 2014.

Vacaflor, P., Lenzano, M. G., Vich, A., and Lenzano, L.: Co-Registration Methods and Error Analysis for Four Decades (1979–2018) of Glacier Elevation Changes in the Southern Patagonian Icefield, Remote Sens., 14(4), 820, https://doi.org/10.3390/rs14040820, 2022.

One example: You write "However, as shown in Section 4, the performance of the NK method is usually suboptimal because only the shift-induced errors are considered. When the precision of DEM co-registration is of great importance, the RT method is recommended for glacial studies to replace the widely used NK method." What does usually mean? Are there cases where the NK performance is optimal, can you citer related studies? The precision is usually of high importace. When would you recommend the RT method? There might be studies to cite in order to show the great importacne of the co-registration. The statement thet RT method is recommended should be repeated in the conclusions.

The word "usually" means "under normal circumstances" or "with high probability". It has been removed in the revised manuscript, because it is not an exact expression. A hypothesis testing approach has been used for deciding whether the precision of the RT method is better than the NK method. The statistical result shows that the null hypothesis cannot be rejected at the $p=0.05$ level.

We recommend the RT method to replace the NK method in glacial studies without any preconditions. This statement is supported by both experimental results (from a statistical viewpoint) and theoretical analysis (the performance of the NK method is limited by ignoring scale- and rotation-induced errors). The RT method has been widely used in photogrammetry and remote sensing studies, but it has not been investigated by the cryosphere community. So, there is no relevant literature to support our conclusions directly.

One example about the vague statement: "Sinusoidal functions usually takes multiple trials, and the performance is sometimes limited by their predefined models." What does "usually" and "sometimes" mean? Here, references to other studies might also help.

The word "usually" has been removed in the revised manuscript. The residual signals induced by satellite attitude jitter often appear at several frequencies, and the numbers are not constant for different DEM sources but depend upon sensors and data pre-processing techniques (before DEM generation). A fixed number of sinusoidal functions cannot be universally suitable for various data sources. A better choice is to fit 1–n (e.g., n=6 in Girod et al., 2017) sinusoidal functions separately and select the best fitting result.

The residual signals in many DEM pairs do have an exact mixed sinusoidal shape, but some do not. The parametric regression method with a predefined model "sometimes" cannot obtain a very high precision fitting result.

**References:**

Girod, L., Nuth, C., Kääb, A., McNabb, R., and Galland, O.: MMASTER: Improved ASTER DEMs for Elevation Change Monitoring, Remote Sens., 9, 704, https://doi.org/10.3390/rs9070704, 2017.

These are two examples but there are more, so please try your best to improve the discussion accordingly. If done propperly I am happy to accept the paper.

Please do not hesitate to ask in case you have any question.

70    I am looking forward to your revised version and promise a rapid handlig of the revision in order to avoind further delays.

Thanks for your detailed review.

---

## Author Response (AR5)

**Response to Editor on "Co-registration and residual correction of digital elevation models: A comparative study"**

Comment received: 18 Oct 2023

5 Key:

Editor comment (black)

Response (blue)
* * *
Thank you for the further improvement. The manuscript has further improved, but I was expecting more effort in
10 improving the discussion. The points I raised were examples and there were more. However, as I do not want to be too
critical and your work clearly deserves publication, I am accepting the manuscript after the following minor issues have been
addressed (Note that the line numbers refer to the track change version):

General comment on the discussion: You provide twice a recommendation to use the RT method (L375 and L382). It
15 would be better to provide one recommendation after you discuss all three methods at the end of the discussion.

Thanks for your suggestion. The sentences have been revised as follows:

To sum up, the three analytical algorithms for the DEM co-registration problem, i.e., NK, NH, and RT, have a strong
theoretical relationship, despite being presented in diverse forms in the original literature. Taking into account both the
algorithm accuracy and ease of use in practical applications, we recommend to apply the RT method instead of the NK and
20 NH methods in glacial studies.

L343: Cite the original publication which introduced the methods (as you did in the intro) and not Paul et al. (2015).
Moreover, clarify the term "grid search method". It was not introduced as such in the introduction.

The literature has been updated according to your suggestion.

25 The grid search method was introduced in Section 1 (Lines 55-58), and we have further elaborated on it in the revised
manuscript.

"The grid search methods search for the best alignment result by stepwise shifting the secondary DEM a little bit,
alternatively along x and y directions in a predefined window (e.g., 5 × 5 pixels). However, these methods have been rarely
used in the recent literature because their brute-force search process comes with a huge computational cost."

30 It is worth noting that there is no universally accepted name for the grid search method in the literature. For instance,
Berthier et al. (2007) referred to it as "planimetric adjustment," while Paul et al. (2015) named it "brute-force iterative
minimisation of difference residuals."

Reference:

35     Berthier, E., Arnaud, Y., Kumar, R., Ahmad, S., Wagnon, P., and Chevallier, P.: Remote sensing estimates of glacier mass balances in the Himachal Pradesh (Western Himalaya, India), Remote Sens. Environ., 108, 327–338, https://doi.org/10.1016/j.rse.2006.11.017, 2007.

Paul, F., Bolch, T., Kääb, A., Nagler, T., Nuth, C., Scharrer, K., Shepherd, A., Strozzi, T., Ticconi, F., Bhambri, R., Berthier, E., Bevan, S., Gourmelen, N., Heid, T., Jeong, S., Kunz, M., Lauknes, T. R., Luckman, A., Merryman Boncori, J.

40   P., Moholdt, G., Muir, A., Neelmeijer, J., Rankl, M., VanLooy, J., and Van Niel, T.: The glaciers climate change initiative: Methods for creating glacier area, elevation change and velocity products, Remote Sens. Environ., 162, 408–426, https://doi.org/10.1016/j.rse.2013.07.043, 2015.

L344: Replace "it" by "therefore the NK method…" (or rewrite the sentence incl. the NH method as suggested above).

45     L375: "replace" is not the best term here. I suggest to write "Based on our theoretical analysis and experimental results, we recommended to apply the RT method instead of the NK method also in glacial studies." (or rewrite the sentence incl. the NH method as suggested above).

L444: Similar as mentioned above: "Thus, we recommend applying the RT method also in glacial studies."

Modified as suggested.

50     Thanks for your detailed review.